# The Batch Complexity of Bandit Pure Exploration

Adrienne Tuynman [1]   Rémy Degenne [1]

## Abstract

In a fixed-confidence pure exploration problem in stochastic multi-armed bandits, an algorithm iteratively samples arms and should stop as early as possible and return the correct answer to a query about the arms distributions. We are interested in batched methods, which change their sampling behaviour only a few times, between batches of observations. We give an instance-dependent lower bound on the number of batches used by any sample efficient algorithm for any pure exploration task. We then give a general batched algorithm and prove upper bounds on its expected sample complexity and batch complexity. We illustrate both lower and upper bounds on best-arm identification and thresholding bandits.

## 1. Introduction

A Multi Armed Bandit (MAB) is a model of a sequential interaction that was introduced in (Thompson, 1933) to create better medical trials. This framework has since been expanded to various fields, and has seen applications to online advertising and recommendation systems. In a MAB, an algorithm chooses at each time an *arm* among a finite number (it *pulls* it) and then observes a sample from a probability distribution associated with the arm. The goal of the interaction will be to identify quickly which arm has the distribution with highest mean.

By making use of past observed rewards to continuously update the way they sample, MAB algorithms reach their objective faster than traditional fixed randomized trials. For applications like online advertising, obtaining feedback can be quick, if for example the feedback is a click on an advertisement. Adapting after almost each interaction is then feasible. However in applications like clinical trials, the delay between giving a treatment to a patient and seeing the effect can be on the order of months. It is not possible to use a fully sequential algorithm, which would take too long. Moreover, adapting at each time step can have a high computational cost.

This motivates looking into *batched* algorithms: the algorithm pulls multiple arms within a single batch, and can only observe the results at the end of the batch. An advantage is that all arms in a batch can be pulled in parallel, reducing greatly the length of the test in applications with large delay. Its lessened adaptivity also reduces the computational cost. Of course, being less adaptive than a typical bandit method could lead to worse algorithms. The key question is then how few batches one can use while keeping a performance comparable to a fully sequential algorithm.

### 1.1. Fixed Confidence Pure Exploration

We study $K$-armed bandit models, which we represent by a vector of real distributions $\boldsymbol{\nu} = (\nu_i)_{i \in [K]}$ with finite means, with the vector of means denoted by $\boldsymbol{\mu} = (\mu_i)_{i \in [K]}$. At each timestep $t \in \mathbb{N}$, an agent interacts with the bandit by choosing an arm $i_t \in [K]$ to sample. It then observes $X_{t,i_t}$ a realization of $\nu_{i_t}$ and proceeds to the next step. One common objective is regret minimization, where the agent must maximize $\mathbb{E}[\sum_t X_{t,i_t}]$. See (Bubeck et al., 2012; Lattimore & Szepesvári, 2020) for extensive surveys. We focus on another goal, pure exploration, in which the agent must answer a question about $\boldsymbol{\nu}$, such as identifying the arm with the highest mean (Even-Dar et al., 2002; Kaufmann et al., 2016). In pure exploration problems, we have a set $\mathcal{I}$ of possible answers, and a function $\boldsymbol{\mu} \mapsto i^\star(\boldsymbol{\mu}) \in \mathcal{I}$ that depends on the means of the distributions and provides the unique correct answer for that instance. For a given pure exploration problem and a vector of means $\boldsymbol{\mu}$, we denote by $Alt_{\boldsymbol{\mu}}$ the set of vectors of means $\boldsymbol{\mu}'$ such that $i^\star(\boldsymbol{\mu}') \neq i^\star(\boldsymbol{\mu})$. Essentially, $Alt_{\boldsymbol{\mu}}$ is the set of means that disagree with $\boldsymbol{\mu}$.

A pure exploration algorithm samples arms in a bandit interaction (following a *sampling rule*) until a time at which it decides to stop (with a *stopping rule*). Once it stops, it returns an answer $\hat{i} \in \mathcal{I}$. A good pure exploration algorithm should have low probability of error $\mathbb{P}_{\boldsymbol{\nu}}\{\hat{i} \neq i^\star(\boldsymbol{\mu})\}$ and stop quickly (use few samples). When the number of samples is fixed and the objective is to minimize the proba-

[1]Univ. Lille, Inria, CNRS, Centrale Lille, UMR 9189-CRIStAL, F-59000 Lille, France. Correspondence to: Adrienne Tuynman <adrienne.tuynman@inria.fr>, Rémy Degenne <remy.degenne@inria.fr>.

*Proceedings of the 42$^{nd}$ International Conference on Machine Learning*, Vancouver, Canada. PMLR 267, 2025. Copyright 2025 by the author(s).

bility of error, one talks about fixed-budget pure exploration (Audibert et al., 2010). We consider instead the *fixed confidence* objective, in which the probability of error is fixed and the agent aims to achieve it with the minimum number of samples. We focus on $\delta$-**correct algorithms**. That is, algorithms that satisfy $\mathbb{P}_{\boldsymbol{\nu}}\{\hat{i} \neq i^{\star}(\boldsymbol{\mu})\} \leq \delta$ (Garivier & Kaufmann, 2016).

We focus on bandit models where each arm $i \in [K]$ has a sub-Gaussian distribution $\nu_i$ of constant $\sigma^2$. That is, for all $\lambda \in \mathbb{R}$, $\mathbb{E}_{\nu_i}[e^{\lambda(X-\mu_i)}] \leq e^{\sigma^2\lambda^2/2}$. For example, both bounded and Gaussian distributions are sub-Gaussian.

Finally, we study **batched** algorithms (Agarwal et al., 2017; Jin et al., 2019). In practical settings, it is not always possible to adapt at every single timestep. Therefore, we want the agent to only observe results and take decisions a limited number of times. At the beginning of a batch, the agent decides how many times to sample each arm $i \in [K]$, then observes all the realizations of those pulls. It then can proceed to the next batch. Crucially, the agent cannot change anything during a batch: it cannot change its sampling decision and cannot stop a batch early. For $\delta$-correct batched algorithms, for each pure exploration problem, we want to control the sample complexity $\tau_\delta$ (the number of samples used), and the batch complexity $R_\delta$ (the number of batches used, which is the number of times the results were observed and the sampling rule was updated).

In all pure exploration problems, the expectation of the stopping time of $\delta$-correct strategies on an instance $\boldsymbol{\nu}$ is bounded from below as follows (Garivier & Kaufmann, 2016). Let $\Sigma_K$ be the simplex, KL the Kullback-Leibler divergence and kl the KL between Bernoulli distributions.

$$\mathbb{E}_{\boldsymbol{\nu}}[\tau_\delta] \geq \tilde{T}^{\star}(\boldsymbol{\mu})\mathrm{kl}(\delta, 1-\delta) \geq \tilde{T}^{\star}(\boldsymbol{\mu})\log(1/(2.4\delta)) \,,$$

in which $(\tilde{T}^{\star}(\boldsymbol{\mu}))^{-1} = \sup_{w \in \Sigma_K} \inf_{\boldsymbol{\lambda} \in Alt_{\boldsymbol{\mu}}} \sum_{i=1}^{K} w_i \mathrm{KL}(\nu_i, \lambda_i) \,.$

This result has been expanded to pure exploration problems with multiple correct answers by (Degenne & Koolen, 2019) in the asymptotic regime $\delta \to 0$. This $\tilde{T}^{\star}(\boldsymbol{\mu})$ is the instance-dependent complexity of instance $\boldsymbol{\mu}$ in this problem. Our objective is to develop algorithms with a sample complexity approaching $\tilde{T}^{\star}(\boldsymbol{\mu}) \log(1/\delta)$ for all $\boldsymbol{\mu}$. As we are in the $\sigma$-subgaussian setting, $\tilde{T}^{\star}(\boldsymbol{\mu}) \leq T^{\star}(\boldsymbol{\mu})$ where

$$(T^{\star}(\boldsymbol{\mu}))^{-1} = \sup_{w \in \Delta_K} \inf_{\boldsymbol{\lambda} \in Alt_{\boldsymbol{\mu}}} \sum_{i} w_i \frac{(\mu_i - \lambda_i)^2}{2\sigma^2}$$

(by Donsker-Varadhan duality, see for example Lemma 2 of (Wang, 2021)) with equality for Gaussians with variance $\sigma^2$. Our goal is to obtain algorithms that have sample complexity as close to $T^{\star}(\boldsymbol{\mu}) \log(1/\delta)$ as possible while using few batches.

We will prove results for general identification problems, and specialize them to three particular cases. Let $\sigma$ be a permutation such that $\mu_{\sigma(1)} \geq \cdots \geq \mu_{\sigma(K)}$.

- In **Best Arm Identification** (BAI), $\mathcal{I} = [K]$ and, for some instance for which $\mu_{\sigma(1)} > \mu_{\sigma(2)}$, the correct answer is $i^{\star}(\boldsymbol{\mu}) = \sigma(1)$ the best arm.

- In **Top-**$k$, $\mathcal{I} = \mathcal{P}_k(K)$ the set of all subsets of $[K]$ of size $k$, and the correct answer for some instance for which $\mu_{\sigma(k)} > \mu_{\sigma(k+1)}$ is $i^{\star}(\boldsymbol{\mu}) = \{\sigma(1), \sigma(2), \ldots, \sigma(k)\}$ the set of the best $k$ arms. Note that top-1 is BAI.

- In the **Thresholding Bandit Problem** (TBP) with threshold $\tau$ (Locatelli et al., 2016), $\mathcal{I}$ is the set of all subsets of $[K]$, and the correct answer for some instance for which $\forall i, \mu_i \neq \tau$ is $i^{\star} = \{i : \mu_i > \tau\}$ the set of all arms that have their mean above the threshold.

## 1.2. Related Work

The problem of batched bandit algorithms has been studied in fixed-confidence BAI and Top-k. Some of the works cited below consider a task with multiple agents that limit the number of rounds in which they communicate, but their methods also give batched bandit algorithms. Hillel et al. (2013) derived an algorithm for $\varepsilon$-BAI (the algorithm needs to return an arm with mean within $\varepsilon$ of the best) which progressively eliminates arms. They show high probability bounds on the batch and sample complexities: with probability $1 - \delta$, it uses less than $\log(1/\varepsilon)$ batches and uses $O\left(\sum_i (\Delta_i^{\varepsilon})^{-2} \log\left(\frac{K}{\delta}\log(\Delta_i^{\varepsilon})^{-1}\right)\right)$ samples with $\Delta_i^{\varepsilon} = \max\{\mu^{\star} - \mu_i, \varepsilon\}$. Agarwal et al. (2017) consider $\delta$-correct algorithms for BAI that use a given number of batches, and give a worst-case upper bound on the number of samples used by their algorithm (also elimination based), but their method requires the knowledge of a lower bound on the gaps of the arms.

Jin et al. (2019) and Karpov et al. (2020) designed $\delta$-correct algorithms for the Top-k problem, and give each bounds with high probability on the batch and sample complexities. (Jin et al., 2023) is the work which is closest to our approach when it comes to algorithms. They study the BAI setting and prove bounds on the expected batch and sample complexity. Compared to bounds with probability $1-\delta$, that requires that they control the complexities also in the event of probability $\delta$. Their method for finite $\delta$ is also elimination based, but they additionally propose an algorithm with guarantees as $\delta \to 0$ that is inspired by the Track-and-Stop BAI algorithm (Garivier & Kaufmann, 2016). Finally, Lu et al. (2019) build a batch-based algorithm for heavy-tailed distributions in a Track-and-Stop fashion, but with the goal of minimizing computational complexity, and thus do not aim to control the batch complexity itself.

All those works cover Top-k or its special case BAI. However, many other pure exploration problems have been studied in the fully sequential setting, like thresholding bandits (Locatelli et al., 2016) or the problem of detecting if any arm has mean larger than a threshold (Kaufmann et al., 2018). We seek to analyze general pure exploration problems and answer the following questions: what is the minimal expected batch sample complexity needed to get an expected sample complexity close to what is achievable by fully sequential algorithms? Can we design a general pure exploration algorithm with near-optimal sample complexity and that reaches that minimal batch complexity?

### 1.3. Contributions

We show a link between sample and batch complexities. We build in Section 2 a general method for computing batch complexities lower bounds for $\delta$-correct pure exploration algorithms. We demonstrate how to apply that method to Top-k (including BAI) and thresholding bandits. The lower bounds we obtain are instance dependent: contrary to previous work, we don't merely state that there exist an instance on which the algorithm requires some number of batches, but we give a lower bound for the batch complexity of each instance, function of its complexity $T^\star$.

In Section 3, we construct a general batched algorithm for pure exploration problems, taking inspiration from Track-and-Stop (Garivier & Kaufmann, 2016). The batch complexity of that algorithm is close to the lower bound under mild conditions that are satisfied on Top-k and thresholding bandits. Moreover, its sample complexity is close to optimal in the high confidence regime (small error probability $\delta$).

## 2. Lower Bound on the Round Complexity

It seems obvious that using more batches will make it possible to use less samples, as the algorithm can more quickly adapt to its observations. What then is the precise relation between the sample complexity and the number of batches?

Tao et al. (2019) focus on the problem of collaborative learning with limited interaction, in which multiple agents take samples in an environment and observe their own rewards, but must minimize the number of times they communicate their rewards with the other agents. They manage to find a link, in some specific 2-armed instances, between how much faster an algorithm can get when communication is allowed, and the number of times the algorithm communicates. They do so by constructing a sequence of gradually more difficult instances, each requiring one more batch than the previous one. Generalizing this idea lets us reach the following result, for any pure-exploration problem.

**Lemma 2.1.** *Suppose that a $\delta$-correct algorithm satisfies $\mathbb{P}_{\boldsymbol{\mu}}(\tau_\delta > \gamma T^\star(\boldsymbol{\mu})\ln(1/\delta)) \leq c$ for some $\gamma, c > 0$ on*

*any Gaussian instance $\boldsymbol{\mu}$ with variance $\sigma^2$ with $T^\star(\boldsymbol{\mu}) \in (T_{\min}, T_{\max})$. Let $(\boldsymbol{\mu}^n)_{0 \leq n \leq N}$ be a sequence of such Gaussian instances with $T^\star(\boldsymbol{\mu}^n) = T^\star(\boldsymbol{\mu}^0)\zeta^{-n} \in (T_{\min}, T_{\max})$ for some $\zeta \in (0,1)$. Then*

$$\mathbb{P}_{\boldsymbol{\mu}^N}(R_\delta > N) \geq 1 - N(2\delta + c) - \sqrt{\frac{\gamma \ln(1/\delta)}{2(\zeta^{-1} - 1)}} S_N$$

*with*

$$S_N = \sum_{n=0}^{N-1} \left[ 1 + \sqrt{\frac{T^\star(\boldsymbol{\mu}^0)}{\zeta^n} \sup_{w \in \Sigma_K} \sum_{i \in [K]} w_i \frac{(\mu_i^{n+1} - \mu_i^n)^2}{2}} \right]$$

In order to get a lower bound on the number of batches, it remains to construct the right instance sequences for each problem. It should start with an easy instance $\boldsymbol{\mu}^0$, and end with $\boldsymbol{\mu}^N = \boldsymbol{\mu}$ the instance whose complexity we want to study, with each succeeding instance being quantifiably harder than the previous one. See an illustration in Figure 1.

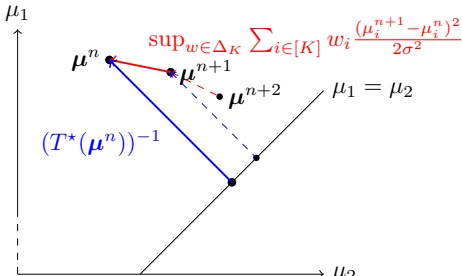

*Figure 1.* Illustration of a sequence of instances

An ideal sequence would minimize the sum $S_N$, thus allowing us to get to a larger $N$. As getting such a minimizing sequence is difficult, we use simpler sequences $\boldsymbol{\mu}^n = \boldsymbol{y} + x^n(\boldsymbol{\mu}^0 - \boldsymbol{y})$ with $\boldsymbol{y}$ the constant vector $(y, y, \ldots, y)$ with $y \in \mathbb{R}$, for some starting mean vector $\boldsymbol{\mu}^0$. In order to be able to do so, an important ingredient is the following condition:

**Assumption 2.2.** For any mean vector $\boldsymbol{\mu}$, there exists $y \in \mathbb{R}$ such that for all $x \in (0,1]$ and $\boldsymbol{\mu}' = x\boldsymbol{\mu} + (1-x)\boldsymbol{y}$ (where $\boldsymbol{y}$ is the constant vector $(y, y, \ldots, y)$), $(T^\star(\boldsymbol{\mu}'))^{-1} = x^2(T^\star(\boldsymbol{\mu}))^{-1}$.

This assumption essentially states that the complexity of the instance scales quadratically as the instance is linearly scaled in that manner, with all arm means concentrating around some value $y$. This is useful for constructing sequences of instances and analyzing how the complexity evolves.

Top-$k$ and thresholding bandits both satisfy that assumption. For top-$k$, concentrating arm means around any value makes arm means get closer, makes gaps shrink linearly, and thus

makes the complexity grow quadratically. For thresholding bandits, concentrating arm means around the threshold value has the same effect. With that condition, it is possible to simplify the bound of Lemma 2.1. We give a general result for the exploration problems satisfying this condition in Appendix A. We then apply this result to Top-k and TBP.

**Theorem 2.3.** *For Top-k and TBP, for a $\delta$-correct algorithm on Gaussian instances with variance $\sigma^2$ of complexity $T^\star \in (T_{\min}, T_{\max})$, we have for any such Gaussian instance $\boldsymbol{\mu}$ of complexity $T^\star(\boldsymbol{\mu}) \in (T_{\min}, T_{\max})$ that*

$$\mathbb{E}_{\boldsymbol{\mu}}[R_\delta] \geq \min \left\{ \frac{\ln \frac{T^\star(\boldsymbol{\mu})}{T_{\min}}}{2 \ln \left( \left( \ln \frac{T^\star(\boldsymbol{\mu})}{T_{\min}} \right)^2 \max\{e, C_\delta\} \right)} , \right.$$

$$\left. \frac{1}{6} \ln \frac{T^\star(\boldsymbol{\mu})}{T_{\min}} , \frac{1}{6\delta} \right\}$$

*with*

$$C_\delta = 1 + 4\gamma \ln \left( \frac{1}{\delta} \right) \ln \frac{T^\star(\boldsymbol{\mu})}{T_{\min}} \left( 1 + \sqrt{\frac{T^\star(\boldsymbol{\mu})\Delta^2}{\sigma^2}} \right)^2 ,$$

$$\gamma = \sup_{\boldsymbol{\nu}: T^\star(\boldsymbol{\mu}) \in (T_{\min}, T_{\max})} \frac{\mathbb{E}_{\boldsymbol{\nu}}[\tau_\delta]}{\ln(1/\delta) T^\star(\boldsymbol{\mu})} .$$

*and where $\Delta = \frac{\max_i \mu_i - \min_i \mu_i}{2}$ in the Top-k setting and $\Delta = \max_i |\mu_i - \tau|$ in thresholding bandits.*

We give all details of the proofs in Appendix A. For $\delta$ small enough, the batch complexity is of order

$$\Omega \left\{ \frac{\ln \frac{T^\star(\boldsymbol{\mu})}{T_{\min}}}{\ln \left( \ln \frac{T^\star(\boldsymbol{\mu})}{T_{\min}} \right) + \ln \left( \gamma \ln(\delta^{-1}) K \frac{\max_i \Delta_i^2}{\min_i \Delta_i^2} \right)} \right\} .$$

This is the first lower bound on batch complexity as a function of the sample complexity for all pure exploration problem satisfying Assumption 2.2. Our algorithm will have a batch complexity of order $\ln \frac{T^\star(\boldsymbol{\mu})}{T_{\min}}$, almost matching the lower bound.

For collaborative bandits (where periods without communication are the analog of batches), Tao et al. (2019) have worked on algorithms $\mathcal{A}_b$ satisfying $\mathbb{E}_{\mathcal{A}_b}[\tau_\delta] = \mathcal{O}\left( \inf_{\mathcal{A}} \mathbb{E}_{\mathcal{A}}[\tau_\delta] \right)$, where the infimum is over fully sequential algorithms. Their results can be adapted to the batched setting to show that there are two-armed instances on which those algorithms must satisfy $\mathbb{E}[R_\delta] = \Omega\left( \frac{\ln \Delta_2^{-1}}{\ln(\ln \Delta_2^{-1})} \right)$. Our result is valid for more pure exploration problems than just BAI; it is a bound on general $K$-armed Gaussian instances rather than specific two-armed instances; and gives instance-dependent bounds instead of merely the existence of an unspecified hard instance.

Theorem 2.3 does not contradict the guarantees of Tri-BBAI (Jin et al., 2023). Their algorithm always uses 3 batches,

but the sample complexity is only controlled in the regime $\delta \to 0$, in which our lower bound goes to 0. Since $\mathbb{E}_{\boldsymbol{\nu}}[\tau_\delta]$ is not controlled otherwise, $\gamma$ can be large, making our lower bound small and consistent with 3 batches.

**Necessity of $T_{\min}$.** The bound depends on $T_{\min}$, which is the minimal complexity for which the sample complexity of the algorithm is bounded by $\gamma T^\star(\boldsymbol{\mu}) \log(1/\delta)$. Since no algorithm uses less than 1 sample, every algorithm has such a $T_{\min}$ greater than $(\log(1/\delta))^{-1}$, and the lower bound is then of order $\log(T^\star(\boldsymbol{\mu}) \log(1/\delta))$. $T_{\min}$ is a tradeoff between prior knowledge and batch complexity. In the extreme situation in which $T^\star(\boldsymbol{\mu})$ is known and the algorithm is $\delta$-correct only on instances of that complexity, the lower bound is 0. Our algorithm will use at most 5 batches in that case: any lower bound has to be a small constant.

## 3. A General Algorithm

We present an algorithm for pure exploration in bandits with upper bounds on both its batch and sample complexities. We take inspiration from the Track-and-Stop method (Garivier & Kaufmann, 2016), which is fully sequential (it does not use batches) and has asymptotically optimal sample complexity. The principle of Track-and-Stop for Gaussians is to try to sample according to $w^\star(\boldsymbol{\mu}) = \text{argmax}_{\boldsymbol{w} \in \Delta_K} \inf_{\boldsymbol{\lambda} \in Alt_{\boldsymbol{\mu}}} \sum_i w_i(\mu_i - \lambda_i)^2/(2\sigma^2)$, the vector of ideal sampling proportions at $\boldsymbol{\mu}$. At time $t$, having sampled arm $i$ approximately $w_i^\star t$ times leads to an algorithm that has minimal asymptotic sample complexity. Of course $w^\star(\boldsymbol{\mu})$ is unknown: Track-and-Stop estimates $\boldsymbol{\mu}$ by its empirical mean $\hat{\boldsymbol{\mu}}$, then approximates $w^\star(\boldsymbol{\mu})$ by $w^\star(\hat{\boldsymbol{\mu}})$ and uses those proportions. Some amount of uniform exploration is added to ensure convergence of the estimates.

We introduce a batched algorithm that first samples uniformly until it has estimated the sampling proportions and complexity well enough, then uses a last phase in which it samples like Track-and-Stop.

### 3.1. Stopping Rule

To check whether we can stop, a commonly used method in parametric pure exploration is the Generalized Likelihood Ratio (GLR) test (Garivier & Kaufmann, 2016). Since we consider the non-parametric class of sub-Gaussian distributions, we use a Gaussian version of that test. The stopping rule of the algorithm is to stop at the end of a phase if

$$\inf_{\boldsymbol{\lambda} \in Alt_{\hat{\boldsymbol{\mu}}^t}} \sum_i N_i^t \frac{(\hat{\mu}_i^t - \lambda_i)^2}{2\sigma^2} > \beta(t, \delta) . \tag{1}$$

**Lemma 3.1** ((Garivier & Kaufmann, 2016))**.** *Any algorithm using the stopping rule (1) with a threshold $\beta(t, \delta)$ satisfying $\mathbb{P}\left( \exists t, \sum_{i=1}^K N_i^t \frac{(\hat{\mu}_i^t - \mu_i)^2}{2\sigma^2} > \beta(t, \delta) \right) \leq \delta$ and returning $i^\star(\hat{\boldsymbol{\mu}}^t)$ is $\delta$-correct.*

We provide below a threshold that works for sub-Gaussian distributions in any pure exploration problem. For particular problems like BAI it should be possible to derive improved thresholds, as was done in parametric cases by Kaufmann & Koolen (2021). Our threshold is derived using the method of mixtures, as in that paper and other bandit articles (Abbasi-Yadkori et al., 2011). It uses the techniques of (Degenne, 2019) to fine-tune the constants. While the BAI literature contains many similar thresholds for parametric settings, we could not find that result for sub-Gaussian distributions.

Let $W_{-1}$ be the negative branch of the Lambert $W$ function and let $\overline{W} : (1, +\infty) \to \mathbb{R}$ be the function defined by $\overline{W}(x) = -W_{-1}(-e^{-x})$. It satisfies $x + \ln x \leq \overline{W}(x) \leq x + \ln x + 1/2 \leq 2x$.

**Lemma 3.2.** *The following threshold satisfies the condition of Lemma 3.1 for $\sigma^2$-sub-Gaussian distributions:*

$$\beta(t, \delta) = \frac{K}{2} \overline{W}\left( \frac{2}{K} \ln \frac{1}{\delta} + 4 \ln\left( \ln \frac{et}{K} \right) + 2 \ln \frac{e\pi^2}{6} \right) .$$

See Appendix B for the proof. Roughly, ignoring additive constants, $\beta(t, \delta) \approx \ln(1/\delta) + 2K \ln \ln t$. The linear dependence in $K$ may be unavoidable in general (for the problem of telling if the means belong to the unit ball, if $\boldsymbol{\mu}$ is the center) but could be improved in problems like BAI. The threshold for exponential families of (Kaufmann & Koolen, 2021) depends logarithmically on the number of arms.

### 3.2. Known Confidence Set

In order to design an algorithm, we first ask the following questions: given enough information, could we return a correct answer within one batch? How would we sample and what would be the sample complexity?

Suppose that we know a set $B$ which likely contains $\boldsymbol{\mu}$ and will likely contain the empirical mean vector $\hat{\boldsymbol{\mu}}_t$ for any amount of sampling we perform. Since we will be using only one batch, we have to select the number of samples for each arm in advance. We want that number of samples to be sufficient to enable the stopping rule, for any $\boldsymbol{\mu}' \in B$. Our solution will be based on two worst-case definitions.

**Definition 3.3.** *For any $B$ a set of instances, define (with $0^{-1} = +\infty$)*

$$\overline{w}^\star(B) = \underset{w \in \Delta_K}{\operatorname{argmax}} \inf_{\boldsymbol{\nu} \in B} \inf_{\boldsymbol{\lambda} \in Alt_{\boldsymbol{\nu}}} \sum_i w_i \frac{(\nu_i - \lambda_i)^2}{2\sigma^2} ,$$

$$\overline{T}^\star(B) = \left( \max_{w \in \Delta_K} \inf_{\boldsymbol{\nu} \in B} \inf_{\boldsymbol{\lambda} \in Alt_{\boldsymbol{\nu}}} \sum_i w_i \frac{(\nu_i - \lambda_i)^2}{2\sigma^2} \right)^{-1} .$$

Note that $\overline{T}^\star(B) \geq \max_{\boldsymbol{\mu} \in B} T^\star(\boldsymbol{\mu})$, but those two quantities may not be equal. To see this, imagine an instance $\boldsymbol{\nu}$ such that $w_i(\boldsymbol{\mu}) \neq w_j(\boldsymbol{\mu})$, and instance $\boldsymbol{\nu}'$ such that arms $i$

and $j$ are switched. Then, $T^\star(\boldsymbol{\mu}) = T^\star(\boldsymbol{\mu}')$, but it is strictly more difficult to sample so that *both* instances are solved at the same time, so $\overline{T}^\star(\{\boldsymbol{\mu}, \boldsymbol{\mu}'\}) > T^\star(\boldsymbol{\mu})$.

**Lemma 3.4.** *If $\overline{T}^\star(B) < +\infty$, then if after sampling each arm $N_i \geq \gamma \overline{w}_i^\star(B) \overline{T}^\star(B)$ times the empirical estimate $\hat{\boldsymbol{\mu}}$ is in $B$, then $\inf_{\boldsymbol{\lambda} \in Alt_{\hat{\boldsymbol{\mu}}}} \sum_i N_i d(\hat{\mu}_i, \lambda_i) \geq \gamma$ .*

*Proof.* By the condition on $N_i$, then the fact that $\hat{\boldsymbol{\mu}} \in B$ and finally definitions of $\overline{w}^\star$ and $\overline{T}^\star$,

$$\inf_{\boldsymbol{\lambda} \in Alt_{\hat{\boldsymbol{\mu}}}} \sum_i N_i d(\hat{\mu}_i, \lambda_i)$$
$$\geq \gamma \overline{T}^\star(B) \inf_{\boldsymbol{\lambda} \in Alt_{\hat{\boldsymbol{\mu}}}} \sum_i \overline{w}_i^\star(B) d(\hat{\mu}_i, \lambda_i)$$
$$\geq \gamma \overline{T}^\star(B) \inf_{\boldsymbol{\nu} \in B} \inf_{\boldsymbol{\lambda} \in Alt_{\boldsymbol{\nu}}} \sum_i \overline{w}_i^\star(B) d(\nu_i, \lambda_i) = \gamma .$$

$\square$

This entails that picking $\gamma \geq \beta(t, \delta)$ allows us to meet the stopping condition after that one batch, if indeed $\hat{\boldsymbol{\mu}} \in B$. The number of samples used in the batch would be $\overline{T}^\star(B)\gamma \geq T^\star(\boldsymbol{\mu})\beta(t, \delta)$. A good set $B$ is therefore a confidence zone for $\hat{\boldsymbol{\mu}}$, and additionally has a small $\frac{\overline{T}^\star(B)}{T^\star(\hat{\boldsymbol{\mu}})}$ ratio to not oversample.

### 3.3. The Algorithm

Our algorithm is shown in pseudo-code in Algorithm 1. It works in phases. Each phase contains at most two batches, although the second batch is rarely done. We define a "phase complexity" $T_r = 2^r T_0$ for $T_0$ a parameter of the algorithm. In each phase $r$, in a first batch we start by sampling arms uniformly until the number of samples for each arm is at least $2^r l_{1,r}$, where $l_{1,r}$ will be set later.

Then we use the empirical mean at that point $\tilde{\boldsymbol{\mu}}^r$ to find a set $B_r$ that likely contains $\mu$. $\hat{B}_r$ is defined as a ball in infinity norm centered at $\tilde{\boldsymbol{\mu}}^r$ and of radius $\varepsilon_r$ which we will choose later. We write $\mathcal{B}_\infty(\tilde{\boldsymbol{\mu}}^r, \varepsilon_r) := \{\boldsymbol{\mu}' \mid \|\tilde{\boldsymbol{\mu}}^r - \boldsymbol{\mu}'\| \leq \varepsilon_r\}$ and set $\hat{B}_r = \mathcal{B}_\infty(\tilde{\boldsymbol{\mu}}^r, \varepsilon_r)$.

We then check whether the worst-case complexity $\overline{T}^\star(B_r)$ exceeds the phase complexity $T_r$. If it does, we skip to the next phase: we need more exploration to get a tighter set. If it does not, we enter the second batch of the phase and sample according to $\gamma_r \overline{w}^\star(B_r) \overline{T}^\star(B_r)$. $\gamma_r$ is chosen large enough such that under a typical event, the algorithm then stops. The expected number of batches will be the number of phases required to reach the right complexity range plus a small constant.

There are several quantities appearing in the algorithm, which we now specify. $N_i$ is the number of samples observed for arm $i$ and $t = \sum_{i=1}^K N_i$. $\beta(t, \delta)$ is the threshold of Lemma 3.2. The uniform exploration amount is

**Algorithm 1** Phased Explore then Track (PET)

1: **Input:** Starting complexity $T_0 \geq 1$
2: $r \leftarrow 0, t \leftarrow 0, \forall i \ N_i \leftarrow 0$
3: **while** not stop **do**
4:     **for** each arm $i \in [K]$ **do**
5:         **while** $N_i < 2^r l_{1,r}$ **do**
6:             Pull arm $i$
7:             $N_i \leftarrow N_i + 1; \ t \leftarrow t+1$
8:         **end while**
9:     **end for**
10:    Compute the empirical mean vector $\tilde{\boldsymbol{\mu}}^r$.
11:    $\hat{B}_r \leftarrow \{\boldsymbol{\mu}' \mid \|\tilde{\boldsymbol{\mu}}^r - \boldsymbol{\mu}'\| \leq \varepsilon_r\}$

$$\overline{w}^\star \leftarrow \underset{w \in \Delta_K}{\operatorname{argmax}} \inf_{\boldsymbol{\nu} \in \hat{B}_r} \inf_{\boldsymbol{\lambda} \in Alt_{\boldsymbol{\nu}}} \sum_i w_i \frac{(\nu_i - \lambda_i)^2}{2\sigma^2} \,,$$

$$\overline{T}^\star(\hat{B}_r)^{-1} \leftarrow \inf_{\boldsymbol{\nu} \in \hat{B}_r} \inf_{\boldsymbol{\lambda} \in Alt_{\boldsymbol{\nu}}} \sum_i \overline{w}_i^\star \frac{(\nu_i - \lambda_i)^2}{2\sigma^2} \,.$$

12:    **if** $\overline{T}^\star(\hat{B}_r) \leq T_r$ **then**
13:       **for** each arm $i \in [K]$ **do**
14:          Pull arm $i$ $\lceil \gamma_r \overline{w}_i^\star \overline{T}^\star \rceil$ times;
15:          $N_i \leftarrow N_i + \lceil \gamma_r \overline{w}_i^\star \overline{T}^\star \rceil; \ t \leftarrow t + \lceil \gamma_r \overline{w}_i^\star \overline{T}^\star \rceil$
16:       **end for**
17:       Compute the empirical mean vector $\hat{\mu}^r$
18:    **end if**
19:    **if** $\inf_{\boldsymbol{\lambda} \in Alt_{\hat{\mu}^r}} \sum_i N_i \frac{(\hat{\mu}_i^r - \lambda_i)^2}{2\sigma^2} > \beta(t, \delta)$ **then**
20:       $i^\star \leftarrow \arg\max_i \hat{\mu}_i^r$
21:       stop $\leftarrow$ true
22:    **end if**
23:    $r \leftarrow r + 1$
24: **end while**

parametrized by $l_{1,r} = 32T_0 \ln(2\sqrt{2K}T_r)$. The confidence region width is $\varepsilon_r = \sqrt{\frac{2\sigma^2}{2^r l_{1,r}} \ln \frac{2K}{p_r}}$ with $p_r = T_{r+1}^{-2}$. Finally, $\gamma_r$ is the solution to $\gamma = \beta(\bar{t}_r, \delta)$, in which $\bar{t}_r = K2^r l_{1,r} + \gamma T_r$. $\bar{t}_r$ is an upper bound for the sample complexity until the end of phase $r$ for any $r$. It is obtained by considering that in the worst case, both batches are entered at each phase up to $r$.

**Theorem 3.5.** *Algorithm 1 is $\delta$-correct.*

This is a consequence of Lemma 3.1 and 3.2.

In order to describe the number of batches and of samples used by the algorithm, we introduce a "good" event, under which the algorithm behaves as expected.

**Definition 3.6.** Let $\mathcal{E}_r$ be the event that in phase $r$, $\hat{\boldsymbol{\mu}}^r$ and $\boldsymbol{\mu}$ belong to $\hat{B}_r$.

That event will happen with high probability and when it does, we can bound the batch and sample complexity.

**Lemma 3.7.** *For any pure exploration problem, PET (Algorithm 1) is such that at phase $r$, if $\mathcal{E}_r$ holds then either*

$\overline{T}^\star(\hat{B}_r) > T_r$ *and the second batch is not entered, or the algorithm stops.*

*Let $R^\star := \min\{r \mid \forall r' \geq r, \ \mathcal{E}_{r'} \implies \overline{T}^\star(\hat{B}_{r'}) \leq T_{r'}\}$ be the first phase after which when the good event happens, the estimated worst-case complexity is small enough. The batch complexity $R_\delta$ and sample complexity $\tau_\delta$ of PET satisfy*

$$R_\delta \leq R^\star + \max\left\{0, R^\star - \left\lceil \log_2 \frac{T^\star(\boldsymbol{\mu})}{T_0} \right\rceil_+ \right\} + 1$$

$$+ 2\sum_{r=1}^{+\infty} \mathbb{I}(\mathcal{E}_r^c)$$

$$\tau_\delta \leq \bar{t}_{R^\star} + \sum_{r=R^\star}^{+\infty} \mathbb{I}(\mathcal{E}_r^c) \bar{t}_{r+1} \,.$$

*where $\lceil x \rceil_+ = \max\{0, \lceil x \rceil\}$.*

Proof in Appendix C.2. The theorem gives upper bounds on the batch and sample complexities that depend on the random variable $R^\star$. We then need to control $R^\star$, which requires an understanding of how $\overline{T}^\star(\hat{B}_r)$ changes as we explore.

### 3.4. Batch and Sample Complexities

We derive upper bounds on the batch and sample complexities of Algorithm 1.

Our algorithm explores uniformly until the worst-case complexity $\overline{T}^\star(\hat{B}_r)$ is close to $T^\star(\boldsymbol{\mu})$. For that to work, the distributions of the arms and the problem geometry need to be such that we can indeed estimate the complexity. To be able to write a bound on the sample and batch complexities, we further need to quantify the speed of estimation. We introduce an assumption to that effect. We will check that this assumption is satisfied on various tasks like best arm identification and thresholding bandits.

**Assumption 3.8.** There exists a function $b : \mathbb{R}^K \mapsto \mathbb{R}_+$ such that for any $\boldsymbol{\mu}$, for all $\varepsilon \leq b(\boldsymbol{\mu})$ and any $\boldsymbol{\mu}' \in \mathcal{B}_\infty(\boldsymbol{\mu}, \varepsilon) := \{\boldsymbol{\nu} \in \mathbb{R}^K \mid \|\boldsymbol{\nu} - \boldsymbol{\mu}\|_\infty \leq \varepsilon\}$, $\ln \overline{T}^\star(\mathcal{B}_\infty(\boldsymbol{\mu}, \varepsilon)) - \ln T^\star(\boldsymbol{\mu}') \leq \varepsilon/b(\boldsymbol{\mu})$ .

Assumption 3.8 allows us to know how much uniform exploration is enough to relate the worst case complexity $\overline{T}^\star(\mathcal{B}_\infty(\boldsymbol{\mu}, \varepsilon))$ over a ball and $T^\star(\boldsymbol{\mu}')$ for $\boldsymbol{\mu}'$ in that ball. We introduce another assumption, which is simpler to check on many problems.

**Assumption 3.9.** For all $\boldsymbol{\mu}$, for all $\varepsilon \geq 0$,

$$\overline{T}^\star(\mathcal{B}_\infty(\boldsymbol{\mu}, \varepsilon)) = \sup_{\boldsymbol{\mu}' \in \mathcal{B}_\infty(\boldsymbol{\mu}, \varepsilon)} T^\star(\boldsymbol{\mu}') \,.$$

The reason for introducing Assumption 3.9 is that it provides a potentially simpler way to prove Assumption 3.8, as we show in the next lemma.

**Lemma 3.10.** *For all $\boldsymbol{\mu}$ and $\boldsymbol{\mu}'$ with $\|\boldsymbol{\mu} - \boldsymbol{\mu}'\|_\infty \leq \sqrt{\sigma^2/(2T^\star(\boldsymbol{\mu}))}$,*

$$|\ln T^\star(\boldsymbol{\mu}') - \ln T^\star(\boldsymbol{\mu})| \leq \sqrt{\frac{8}{\sigma^2} T^\star(\boldsymbol{\mu})} \, \|\boldsymbol{\mu} - \boldsymbol{\mu}'\|_\infty \, .$$

*As a consequence, if Assumption 3.9 is satisfied, then Assumption 3.8 is satisfied with $b(\boldsymbol{\mu}) = \sqrt{\sigma^2/(8T^\star(\boldsymbol{\mu}))}$.*

We can now present the guarantees of our algorithm, for any pure exploration problem for which Assumption 3.8 holds.

**Theorem 3.11.** *Suppose that Assumption 3.8 is satisfied and let $T_b^\star(\boldsymbol{\mu}) := \max\left\{\frac{\sigma^2}{b(\boldsymbol{\mu})^2}, 2eT^\star(\boldsymbol{\mu})\right\}$. Then PET (Algorithm 1) has expected batch and sample complexities which satisfy*

$$\mathbb{E}\left[R_\delta\right] \leq \max\left\{\log_2 \frac{T_b^\star(\boldsymbol{\mu})}{T_0} + \log_2 \frac{T_b^\star(\boldsymbol{\mu})}{T^\star(\boldsymbol{\mu})}, 0\right\} + 2 \, ,$$

$$\begin{aligned}
\mathbb{E}\left[\tau_\delta\right] \leq{}& 4\ln\left(\frac{1}{\delta}\right)\left(\max\{T_0, T_b^\star(\boldsymbol{\mu})\} + T_0^{-1}\right) \\
&+ 20K(\ln K + 4)(\max\{T_0, T_b^\star(\boldsymbol{\mu})\} + T_0^{-1}) \\
&+ 48K\big(\max\{T_0, T_b^\star(\boldsymbol{\mu})\}\ln\max\{T_0, T_b^\star(\boldsymbol{\mu})\} \\
&+ T_0^{-1}\ln(4T_0)\big) \, .
\end{aligned}$$

If Assumption 3.9 is satisfied, we get from Lemma 3.10 that $T_b^\star(\boldsymbol{\mu}) \leq 8T^\star(\boldsymbol{\mu})$. The batch complexity of the algorithm is then bounded by $\log_2 \frac{T^\star(\boldsymbol{\mu})}{T_0} + 5$. This should be compared to the $\ln \frac{T^\star(\boldsymbol{\mu})}{T_{\min}}$ term of the lower bound, where $T_{\min}$ is the smallest complexity on which the algorithm is both $\delta$-correct and has expected sample complexity close to $T^\star(\boldsymbol{\mu})\log(1/\delta)$. Since PET uses $T_0$ for the first guess of the sample complexity, it cannot match the sample complexity of an instance $\boldsymbol{\mu}$ with $T^\star(\boldsymbol{\mu}) \leq T_0$, hence $T_0$ is the $T_{\min}$ of our algorithm. Hence, PET matches the $\ln \frac{T^\star(\boldsymbol{\mu})}{T_{\min}}$ component of the batch complexity lower bound. We also remark that if we know exactly $T^\star(\boldsymbol{\mu})$ in advance, then we can use it as $T_0$ and the algorithm runs in at most 5 batches.

On the sample complexity side, the $\delta$-dependent term is proportional to $T^\star(\boldsymbol{\mu})\log(1/\delta)$, which is the right dependence in $\delta$, up to the multiplicative constant. The optimal asymptotic complexity as $\delta \to 0$ is exactly $T^\star(\boldsymbol{\mu})\log(1/\delta)$, with factor 1. Our constant could be improved: we made the bound simple at the expanse of a few larger constants. For example the factor 4 in $4\ln\left(\frac{1}{\delta}\right)\left(T_b^\star(\boldsymbol{\mu}) + T_0^{-1}\right)$ is the result of using the coarse inequality $x + \log x \leq 2x$ twice. In the definition of $T_b^\star(\boldsymbol{\mu})$, the 2 in $2eT^\star(\boldsymbol{\mu})$ is due to the choice of doubling $T_r$ at each phase: choosing a multiplication by $(1 + \varepsilon)$ instead of 2 would reduce that. The $e$ factor, as well as the 8 in $b(\boldsymbol{\mu})$, could also be reduced to $1 + \varepsilon$ at the cost of constant terms elsewhere.

The dominant term of the sample complexity as function of $T^\star(\boldsymbol{\mu})$ is $48KT^\star(\boldsymbol{\mu})\ln T^\star(\boldsymbol{\mu})$. Jamieson et al. (2014)

have shown that for two arms the optimal dependence is $O(T^\star(\boldsymbol{\mu})\ln\ln T^\star(\boldsymbol{\mu}))$, which means that our algorithm loses a factor $K$. This is due to the uniform exploration: we sample until every arm's mean is estimated within $\sqrt{T^\star(\boldsymbol{\mu})/\sigma^2}$. We conjecture that a more adaptive exploration could improve that dependence. It is possible in BAI, as demonstrated by Jin et al. (2023). How to do it for any pure exploration problem is an open question.

**Sketch of proof** By Lemma 3.7, we can bound the batch and sample complexities by finding an upper bound for $R^* = \min\{r \mid \forall r' \geq r, \, \mathcal{E}_{r'} \implies \overline{T}^\star(\hat{B}_{r'}) \leq T_{r'}\}$ and by bounding the probability of $\mathcal{E}_r$.

Let $r_0 = \min\{r \mid 2\varepsilon_r \leq b(\boldsymbol{\mu})\}$, where $b(\boldsymbol{\mu})$ is defined in Assumption 3.8. Then for any $r \geq r_0$, under $\mathcal{E}_r$, $\hat{B}_r = \mathcal{B}_\infty(\tilde{\boldsymbol{\mu}}_r, \varepsilon_r) \subseteq \mathcal{B}_\infty(\boldsymbol{\mu}, 2\varepsilon_{r_0})$ and thus for any $\boldsymbol{\mu}' \in \hat{B}_r$, $\ln \overline{T}^\star(\hat{B}_r) - \ln T^\star(\boldsymbol{\mu}') \leq 1$. Hence in the event $\mathcal{E}_r$, we get $\overline{T}^\star(\hat{B}_r) \leq eT^\star(\boldsymbol{\mu})$ since $\boldsymbol{\mu} \in \hat{B}_r$.

Let $r_1 = \min\{r \mid T_r \geq eT^\star(\boldsymbol{\mu})\}$. Then for $r \geq \max\{r_0, r_1\}$, in the event $\mathcal{E}_r$, $T_r \geq eT^\star(\boldsymbol{\mu}) \geq \overline{T}^\star(\hat{B}_r)$. We get that $R^* \leq r^* := \max\{r_0, r_1\}$.

By concentration, since we suppose $\sigma^2$-sub-Gaussian arm distributions, for $\varepsilon_r = \sqrt{\frac{2\sigma^2}{2^r l_{1,r}} \log \frac{2K}{p_r}}$ we have $\mathbb{P}(\mathcal{E}_r) \leq p_r$. We can thus bound the expected batch and sample complexities.

$$\begin{aligned}
\mathbb{E}\left[R_\delta\right] \leq{}& r^* + \max\left\{0, r^* - \left\lceil \log_2 \frac{T^\star(\boldsymbol{\mu})}{T_0}\right\rceil_+\right\} + 1 \\
&+ 2\sum_{r=1}^{+\infty} p_r \, ,
\end{aligned}$$

$$\mathbb{E}\left[\tau_\delta\right] \leq \bar{t}_{r^*} + \sum_{r=1}^{+\infty} p_r \bar{t}_{r+1} \, .$$

With our choices of $p_r$, $\gamma_r$ and $l_{1,r}$, we can finally compute bounds on the sums, $r_0$ and $r_1$ (see Appendix C.2).

### 3.5. Best Arm Identification and Thresholding Bandits

We have provided a general theorem about Algorithm 1 on a generic pure exploration task with sub-Gaussian distributions. However, that theorem requires that Assumption 3.8 be satisfied. We now show that Assumption 3.8 holds on the Top-k task (including BAI) and on the thresholding bandit problem, by showing that we have Assumption 3.9.

**Lemma 3.12.** *In Top-k, including best arm identification, as well as for thresholding bandits, Assumption 3.9 holds. That is, for all $\boldsymbol{\mu}$ and all $\varepsilon \geq 0$,*

$$\overline{T}^\star(\mathcal{B}_\infty(\boldsymbol{\mu}, \varepsilon)) = \max_{\boldsymbol{\mu}' \in \mathcal{B}_\infty(\boldsymbol{\mu}, \varepsilon)} T^\star(\boldsymbol{\mu}') \, .$$

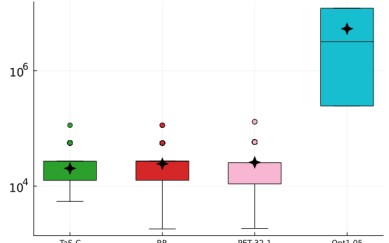

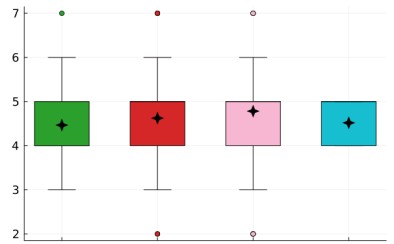

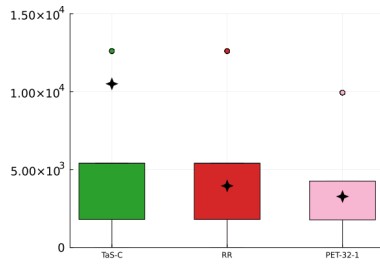

(a) Number of samples before stopping in a random BAI instance, logarithmic scale

(b) Number of rounds before stopping in a random BAI instance

(c) Number of samples before stopping in the min. threshold setting, hard instance

Figure 2. Experimental results, $\delta = 0.05$, $N = 1000$ runs

The proof (in Appendix C.5) hinges on the fact that, whenever it contains only one answer, $\mathcal{B}_\infty(\boldsymbol{\mu}, \varepsilon)$ contains one instance that is more difficult than all the others, in the sense that sampling optimally to decide the answer of that instance also solves all other instances in the set. This is a stronger property than the equality of the lemma. From the existence of that hardest instance, we also get a computationally easy way to compute $\overline{w}^\star(\mathcal{B}_\infty(\boldsymbol{\mu}, \varepsilon))$ and $\overline{T}^\star(\mathcal{B}_\infty(\boldsymbol{\mu}, \varepsilon))$: first compute $\boldsymbol{b}$, and then compute $w^\star(\boldsymbol{b})$ and $T^\star(\boldsymbol{b})$.

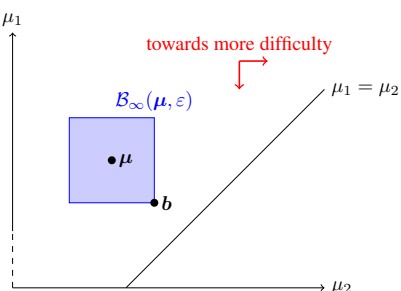

Figure 3. $\boldsymbol{b}$ satisfying $\overline{T}^\star(\mathcal{B}_\infty(\boldsymbol{\mu}, \varepsilon)) = T^\star(\boldsymbol{b})$

We display in Figure 3 a representation of what happens in two-armed BAI. As long as $\mu_1 > \mu_2$, bringing $\mu_1$ down and $\mu_2$ up can only make the problem harder. If $\varepsilon$ is small enough that all the instances in $\mathcal{B}_\infty(\boldsymbol{\mu}, \varepsilon)$ have the same answer, then the instance on the corner of $\mathcal{B}_\infty(\boldsymbol{\mu}, \varepsilon)$ closest to the line $\mu_1 = \mu_2$ is strictly more difficult than all the others. That means that sampling enough to solve it is enough to solve all the other instances.

### 3.6. Experiments on the BAI setting

Our algorithm PET is near-optimal in round and sample complexities for many pure exploration problems, and has theoretical guarantees for any pure exploration problem. To ascertain its practical performances, we compare it to baselines and state of the art algorithms for best arm identification and thresholding bandits.

Each experiment is repeated over 1000 runs. All reward

distributions are Gaussian with variance 1 and we use the confidence level $\delta = 0.05$, which is chosen for its relevance to statistical practice. We compare

- Round Robin (or uniform sampling), where the stopping rule is checked only at timesteps $(900 \times 2^r)_{r \geq 1}$;
- Track-and-Stop (TaS) (Garivier & Kaufmann, 2016), where the empirical value of $w$ is updated only at timesteps $(900 \times 2^r)_r$, and the stopping rule is only checked at those times;
- Our algorithm PET, with $T_0 = 1$;
- Opt-BBAI (Jin et al., 2023) with $\alpha = 1.05$ and the quantities described in their Theorem 4.2.

The initial batch sizes for TaS and Round Robin were chosen to approximate the initial batch size of our algorithm, to not disadvantage them in terms of round complexity. We modified TaS in order to turn it into a batch algorithm. Note that there is no formal guarantee for the batch or sample complexity of that modification of TaS, but we use it as a sensible baseline.

For the BAI experiment, we run each algorithm on 10-arm instances where the best arm has mean 1, and each other arm $i$ has mean uniformly sampled between 0.6 and 0.9. See Figure 2(a) for the box plots of the sample complexities. The mean is indicated by a black cross. While both our algorithm and Opt-BBAI use similarly few batches, PET outperforms Opt-BBAI for the sample complexity. That algorithm is asymptotically optimal as $\delta \to 0$ but it uses batches that seem to be too large for moderate values of $\delta$ like the 0.05 we use.

While the batch modification of TaS might seem to be a good alternative for the BAI experiment, there are instances of the thresholding setting where it performs sub-optimally. That effect was first observed in (Degenne et al., 2019) for the fully online TaS and reflects that, contrary to our results, the sample complexity guarantees of TaS are only asymptotic. We run the algorithms on a thresholding bandit with threshold 0.6 and two arms with means 0.5 and 0.6 and ob-

serve that batched TaS has high average sample complexity (see Figure 2(c); the mean is the black cross), while PET does not.

## 4. Perspectives

We proved instance-dependent lower bounds for the batch complexity of any $\delta$-correct pure exploration algorithm. These lower bounds get larger as the sample complexity of the algorithm decreases. We introduced the Phased Explore then Track algorithm, for which we proved an upper bound for the sample complexity close to the lower bound for fully adaptive methods, as well as an upper bound for the batch complexity that is close to our lower bound.

The main open question raised by our work is how to explore in a better way than uniformly for a general pure exploration problem. The goal of the exploration is to find a set $\hat{B}_r$ such that the worst case complexity $\overline{T}^\star(\hat{B}_r)$ is close to $T^\star(\boldsymbol{\mu})$, which means estimating both $T^\star(\boldsymbol{\mu})$ and $w^\star(\boldsymbol{\mu})$. Uniform exploration leads to a close to optimal batch complexity and to a sample complexity which has a good dependence in $\log(1/\delta)$. However, it comes at the cost of $KT^\star(\boldsymbol{\mu})$ samples used to explore every arm, which should ideally be around $T^\star(\boldsymbol{\mu})$ instead. For BAI and Top-k, the elimination strategies of (Hillel et al., 2013; Jin et al., 2023) achieve that improvement, but the elimination criterion uses the particular link between the gaps and $T^\star$ in Top-k. We would need to find a way to extend the elimination approach to other problems, for which there might not even be an obvious notion of gaps.

## Acknowledgements

The authors acknowledge the funding of the French National Research Agency under the project FATE (ANR22-CE23-0016-01) and the PEPR IA FOUNDRY project (ANR-23-PEIA-0003). The authors are members of the Inria team Scool.

## Impact statement

This paper presents work whose goal is to advance the field of Machine Learning. There are many potential societal consequences of our work, none which we feel must be specifically highlighted here.

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

# A. Proofs of the lower bounds

## A.1. Preliminary lemmas

For the sake of completeness, we start by restating and proving some results from (Tao et al., 2019) in slightly more general language.

**Definition A.1.** For some integers $r$ and $n$, define $\tau_\delta^r$ the number of samples before the end of round $r$.

**Lemma A.2** (Generalization of Lemma 27 of (Tao et al., 2019)). *For an algorithm, two instances $\nu$ and $\nu'$ and $r \in \mathbb{N}$,*

$$\mathbb{P}_{\nu'}\{R_\delta \geq r+1, \tau_\delta^{r+1} \leq n+m\} \geq \mathbb{P}_\nu\{R_\delta \geq r+1, \tau_\delta^r \leq m\} - \mathbb{P}_\nu\{\tau_\delta > n\} - \|\mathcal{D}_\nu^m - \mathcal{D}_{\nu'}^m\|_{TV}$$

*where $\mathcal{D}_\nu^m$ is the distribution of rewards the algorithm got from $\nu$ over $m$ steps.*

*Proof.* Fix a deterministic algorithm.

First of all,

$$(R_\delta \geq r+1, \tau_\delta^r \leq m, \tau_\delta^{r+1} - \tau_\delta^r < n) \subseteq (R_\delta \geq r+1, \tau_\delta^{r+1} \leq n+m) \tag{2}$$

And, since $(R_\delta \geq r+1, \tau_\delta^r \leq m, \tau_\delta^{r+1} - \tau_\delta^r < n)$ is determined by the first $m$ rewards (at the end of round $r$ using less than $m$ samples, the algorithm must choose the length of round $r+1$),

$$\mathbb{P}_{\nu'}\{R_\delta \geq r+1, \tau_\delta^r \leq m, \tau_\delta^{r+1} - \tau_\delta^r < n\} \geq \mathbb{P}_\nu\{R_\delta \geq r+1, \tau_\delta^r \leq m, \tau_\delta^{r+1} - \tau_\delta^r < n\} - \|\mathcal{D}_\nu^m - \mathcal{D}_{\nu'}^m\|_{TV} \tag{3}$$

On the other hand,

$$(R_\delta \geq r+1, \tau_\delta^r \leq m) \setminus (R_\delta \geq r+1, \tau_\delta^r \leq m, \tau_\delta^{r+1} - \tau_\delta^r < n) = (R_\delta \geq r+1, \tau_\delta^r \leq m, \tau_\delta^{r+1} - \tau_\delta^r \geq n)$$
$$\subseteq (\tau_\delta > n)$$

hence

$$\mathbb{P}_\nu\{R_\delta \geq r+1, \tau_\delta^r \leq m, \tau_\delta^{r+1} - \tau_\delta^r < n\} \geq \mathbb{P}_\nu\{R_\delta \geq r+1, \tau_\delta^r \leq m\} - \mathbb{P}_\nu\{\tau_\delta > n\} \tag{4}$$

Hence, using Equations (2), (3) then (4),

$$\begin{aligned}
\mathbb{P}_{\nu'}\{R_\delta \geq r+1, \tau_\delta^{r+1} \leq n+m\} &\geq \mathbb{P}_{\nu'}\{R_\delta \geq r+1, \tau_\delta^r \leq m, \tau_\delta^{r+1} - \tau_\delta^r < n\} \\
&\geq \mathbb{P}_\nu\{R_\delta \geq r+1, \tau_\delta^r \leq m, \tau_\delta^{r+1} - \tau_\delta^r < n\} - \|\mathcal{D}_\nu^m - \mathcal{D}_{\nu'}^m\|_{TV} \\
&\geq \mathbb{P}_\nu\{R_\delta \geq r+1, \tau_\delta^r \leq m\} - \mathbb{P}_\nu\{\tau_\delta > n\} - \|\mathcal{D}_\nu^m - \mathcal{D}_{\nu'}^m\|_{TV}
\end{aligned}$$

$\square$

**Lemma A.3** (Generalization of Lemma 26 of (Tao et al., 2019)). *For any $\delta$-correct algorithm, for all $m, r \in \mathbb{N}$ and any two bandit instances $\nu, \nu'$, we have*

$$\mathbb{P}_\nu\{R_\delta \geq r+1, \tau_\delta^r \leq m\} \geq \mathbb{P}_\nu\{R_\delta \geq r, \tau_\delta^r \leq m\} - 2\delta - \|\mathcal{D}_\nu^m - \mathcal{D}_{\nu'}^m\|_{TV} .$$

*Proof.* Consider the event $\mathcal{F}_1 = (R_\delta = r, \tau_\delta \leq m)$. Denote by $\mathcal{F}_2$ the event that the algorithm returns the best arm of instance $\nu$. Then $\mathbb{P}_\nu\{\mathcal{F}_1\} = \mathbb{P}_\nu\{\mathcal{F}_1 \wedge \mathcal{F}_2\} + \mathbb{P}_\nu\{\mathcal{F}_1 \wedge \overline{\mathcal{F}_2}\}$

With $\mathcal{D}_\nu^m$ the distribution of rewards over $m$ samples and some $\nu' \in Alt_\nu$,

$$\begin{aligned}
\mathbb{P}_\nu\{\mathcal{F}_1 \wedge \mathcal{F}_2\} &\leq \mathbb{P}_{\nu'}\{\mathcal{F}_1 \wedge \mathcal{F}_2\} + \|\mathcal{D}_\nu^m - \mathcal{D}_{\nu'}^m\|_{TV} \\
&\leq \mathbb{P}_{\nu'}\{\mathcal{F}_2\} + \|\mathcal{D}_\nu^m - \mathcal{D}_{\nu'}^m\|_{TV} \\
&\leq \delta + \|\mathcal{D}_\nu^m - \mathcal{D}_{\nu'}^m\|_{TV} .
\end{aligned}$$

On the other hand, $\mathbb{P}_\nu\{\mathcal{F}_1 \wedge \overline{\mathcal{F}_2}\} \leq \mathbb{P}_\nu\{\overline{\mathcal{F}_2}\} \leq \delta$. Therefore $\mathbb{P}_\nu\{\mathcal{F}_1\} \leq 2\delta + \|\mathcal{D}_\nu^m - \mathcal{D}_{\nu'}^m\|_{TV}$. Using $\mathbb{P}_\nu\{R_\delta \geq r+1, \tau_\delta^r \leq m\} \geq \mathbb{P}_\nu\{R_\delta \geq r, \tau_\delta^r \leq m\} - \mathbb{P}_\nu(\mathcal{F}_1)$, we conclude. $\square$

**Lemma A.4.** *For any $\delta$-correct algorithm, for all $m, r \in \mathbb{N}$ and any bandit instance $\boldsymbol{\nu}$, we have*

$$\mathbb{P}_{\boldsymbol{\nu}}\{R_\delta \geq r + 1, \tau_\delta^r \leq m\} \geq \mathbb{P}_{\boldsymbol{\nu}}\{R_\delta \geq r, \tau_\delta^r \leq m\} - 2\delta - \sqrt{\frac{m}{2}(T^\star(\boldsymbol{\nu}))^{-1}}\,.$$

*Proof.* First apply Lemma A.3 to an arbitrary instance $\boldsymbol{\nu}' \in Alt_{\boldsymbol{\nu}}$. Then using Pinsker's inequality yields

$$\|\mathcal{D}_{\boldsymbol{\nu}}^m - \mathcal{D}_{\boldsymbol{\nu}'}^m\|_{TV} \leq \sqrt{\frac{1}{2}\text{KL}(\mathcal{D}_{\boldsymbol{\nu}}^m\|\mathcal{D}_{\boldsymbol{\nu}'}^m)} = \sqrt{\frac{1}{2}\sum_{i\in[K]}\mathbb{E}_{\boldsymbol{\nu}}[N_{m,i}]\frac{(\mu_i - \mu_i')^2}{2}}$$

with $N_{m,i}$ the number of times arm $i$ is pulled before time $m$.

As this is true for all instances $\boldsymbol{\nu}' \in Alt_{\boldsymbol{\nu}}$, we can obtain an inequality using the infimum over those instances,

$$\inf_{\boldsymbol{\nu}'\in Alt_{\boldsymbol{\nu}}}\|\mathcal{D}_{\boldsymbol{\nu}}^m - \mathcal{D}_{\boldsymbol{\nu}'}^m\|_{TV} \leq \sqrt{\frac{m}{2}\inf_{\boldsymbol{\lambda}\in Alt_{\boldsymbol{\nu}}}\sum_{i\in[K]}\frac{\mathbb{E}_{\boldsymbol{\nu}}[N_{m,i}]}{m}\frac{(\mu_i - \lambda_i)^2}{2}}$$

$$\leq \sqrt{\frac{m}{2}\sup_{w\in\Sigma_K}\inf_{\boldsymbol{\lambda}\in Alt_{\boldsymbol{\nu}}}\sum_{i\in[K]}w_i\frac{(\mu_i - \lambda_i)^2}{2}}$$

$$= \sqrt{\frac{m}{2}(T^\star(\boldsymbol{\nu}))^{-1}}\,,$$

by definition of $T^\star$. □

Finally, we also give a technical result to solve inequalities of the form $(k + N^2(a + b\ln N))^N \leq \rho$.

**Lemma A.5.** *Let $\rho \geq e$, $a, b \geq 0$ and $k$ be real numbers, and let $A = \max\{e, k + a\}$. Then $N := \left\lfloor \frac{\ln\rho}{\ln((\ln\rho)^2(A+b\ln\ln\rho))} \right\rfloor$ satisfies $(k + N^2(a + b\ln N))^N \leq \rho$.*

*Proof.* If $N = 0$, the equality is $1 \leq \rho$, which is true since $\rho \geq e$. Otherwise, $N \geq 1$ and $(\ln\rho)^2(A + b\ln\ln\rho) \geq A \geq e$, so $N \leq \lfloor\ln\rho/\ln e\rfloor \leq \ln\rho$. Therefore

$$N\ln(k + N^2(a + b\ln N)) \leq N\ln(N^2(A + b\ln N))$$
$$\leq N\ln((\ln\rho)^2(A + b\ln\ln\rho))$$
$$\leq \ln\rho$$

and finally $(k + N^2(a + b\ln N))^N \leq \rho$. □

## A.2. The lower bound in the general cases

We give here a result for any sequence of instances.

**Lemma A.6.** *Let there be a sequence of instances $(\boldsymbol{\nu}^n)_{0\leq n\leq N}$ such that the probability of error is bounded by $\delta$ and for any $n \in [0, N-1]$, $c_n \geq \mathbb{P}_{\boldsymbol{\nu}^n}[\tau_\delta > x_n]$. Then*

$$\mathbb{P}_{\boldsymbol{\nu}^N}[R_\delta > N] \geq 1 - 2N\delta - \sum_{i=0}^{N-1}\left[c_n + \sqrt{\frac{X_{n-1}}{2}}\left(\sqrt{\frac{1}{T^\star(\boldsymbol{\mu}^n)}} + \sqrt{\sum_{i\in[K]}\frac{\mathbb{E}[N_{X_{n-1},i}]}{X_{n-1}}\frac{(\mu_i^{n+1} - \mu_i^n)^2}{2\sigma^2}}\right)\right]$$

*where $X_n = \sum_{i=-1}^n x_i$, $x_{-1}$ is any positive real number, and $N_{t,i}$ is the number of times arm $i$ is sampled before time $t$.*

*Proof of Lemma A.6.* By lemmas A.2 and A.4, for any $m$,

$$\mathbb{P}_{\boldsymbol{\nu}^{n+1}}\{R_\delta \geq n+1, \tau_\delta^{n+1} \leq m+x_n\} \geq \mathbb{P}_{\boldsymbol{\nu}^n}\{R_\delta \geq n+1, \tau_\delta^n \leq m\} - c_n - \|\mathcal{D}_{\boldsymbol{\nu}^n}^m - \mathcal{D}_{\boldsymbol{\nu}^{n+1}}^m\|_{TV}$$

$$\geq \mathbb{P}_{\boldsymbol{\nu}^n}\{R_\delta \geq n, \tau_\delta^n \leq m\} - 2\delta - \sqrt{\frac{m}{2}(T^\star(\boldsymbol{\mu}^n))^{-1}}$$

$$- c_n - \sqrt{\frac{1}{2}\sum_{i \in [K]} \mathbb{E}[N_{m,i}]\frac{(\mu_i^{n+1} - \mu_i^n)^2}{2\sigma^2}}$$

and with $X_n = \sum_{i=-1}^n x_i$,

$$\mathbb{P}_{\boldsymbol{\nu}^{n+1}}\{R_\delta \geq n+1, \tau_\delta^{n+1} \leq X_n\} \geq \mathbb{P}_{\boldsymbol{\nu}^n}\{R_\delta \geq n, \tau_\delta^n \leq X_{n-1}\} - 2\delta - c_n - \sqrt{\frac{X_{n-1}}{2}(T^\star(\boldsymbol{\mu}))^{-1}}$$

$$- \sqrt{\frac{X_{n-1}}{2}\sum_{i \in [K]} \frac{\mathbb{E}[N_{X_{n-1},i}]}{X_{n-1}}\frac{(\mu_i^{n+1} - \mu_i^n)^2}{2\sigma^2}}$$

So that finally

$$\mathbb{P}_{\boldsymbol{\nu}^N}\{R_\delta \geq N, \tau_\delta^N \leq X_{N-1}\} \geq \mathbb{P}_{\boldsymbol{\nu}^0}\{R_\delta \geq 0, \tau_\delta^0 \leq x_{-1}\} - 2N\delta$$

$$- \sum_{i=0}^{N-1}\left[c_n + \sqrt{\frac{X_{n-1}}{2}}\left(\sqrt{(T^\star(\boldsymbol{\mu}^n))^{-1}} + \sqrt{\sum_{i \in [K]}\frac{\mathbb{E}[N_{X_{n-1},i}]}{X_{n-1}}\frac{(\mu_i^{n+1} - \mu_i^n)^2}{2\sigma^2}}\right)\right]$$

and we conclude since for any $x_{-1} \geq 0$, $\mathbb{P}_{\boldsymbol{\nu}^0}\{R_\delta \geq 1, \tau_\delta^0 \leq x_{-1}\} = 1$ (we always use at least 1 round). $\qquad\square$

From there, we derive the result for $T^\star(\boldsymbol{\mu}^n) = \zeta^{-n}T^\star(\boldsymbol{\mu}^0)$.

**Lemma 2.1.** *Suppose that a $\delta$-correct algorithm satisfies $\mathbb{P}_{\boldsymbol{\mu}}(\tau_\delta > \gamma T^\star(\boldsymbol{\mu})\ln(1/\delta)) \leq c$ for some $\gamma, c > 0$ on any Gaussian instance $\boldsymbol{\mu}$ with variance $\sigma^2$ with $T^\star(\boldsymbol{\mu}) \in (T_{\min}, T_{\max})$. Let $(\boldsymbol{\mu}^n)_{0 \leq n \leq N}$ be a sequence of such Gaussian instances with $T^\star(\boldsymbol{\mu}^n) = T^\star(\boldsymbol{\mu}^0)\zeta^{-n} \in (T_{\min}, T_{\max})$ for some $\zeta \in (0,1)$. Then*

$$\mathbb{P}_{\boldsymbol{\mu}^N}(R_\delta > N) \geq 1 - N(2\delta + c) - \sqrt{\frac{\gamma\ln(1/\delta)}{2(\zeta^{-1} - 1)}}S_N$$

*with*

$$S_N = \sum_{n=0}^{N-1}\left[1 + \sqrt{\frac{T^\star(\boldsymbol{\mu}^0)}{\zeta^n}\sup_{w \in \Sigma_K}\sum_{i \in [K]}w_i\frac{(\mu_i^{n+1} - \mu_i^n)^2}{2}}\right]$$

*Proof of Lemma 2.1.* We apply Lemma A.6 on the sequence $(\boldsymbol{\nu}^n)_{0 \leq n \leq N}$ with $x_{-1} = \gamma T^\star(\boldsymbol{\mu}^0)\log(1/\delta)\frac{1}{\zeta^{-1}-1}$. That way,

$$X_n = x_{-1} + \sum_{i=0}^n \gamma T^\star(\boldsymbol{\mu}^i)\log(1/\delta)$$

$$= \gamma T^\star(\boldsymbol{\mu}^0)\log(1/\delta)\left(\frac{1}{\zeta^{-1}-1} + \sum_{i=0}^n \zeta^{-i}\right)$$

$$= \gamma T^\star(\boldsymbol{\mu}^0)\log(1/\delta)\frac{\zeta^{-(n+1)}}{\zeta^{-1} - 1}$$

$\qquad\square$

Under Assumption 2.2, we can pick a sequence of instances of means $\boldsymbol{\mu}^{n+1} = x\boldsymbol{\mu}^n + (1-x)\boldsymbol{y}$ and control the sequence of $T^\star(\boldsymbol{\mu}^n)$. That way, we get the following result:

**Lemma A.7** (Batch lower bound on affine sequences). *For problems on which Assumption 2.2 is satisfied; for any algorithm such that, for any Gaussian instance $\nu$ satisfying $T^\star(\mu) \in (T_{\min}, T_{\max})$ the probability of error is smaller than $\delta$ and such that $\mathbb{P}_\nu(\tau_\delta > \gamma \log(1/\delta)T^\star(\mu)) \leq c$; we have for any $\sigma$-Gaussian instance $\nu$ of complexity $T^\star(\mu) \in (T_{\min}, T_{\max})$, for the corresponding $y \in \mathbb{R}$ given by Assumption 2.2 for $\mu$, that $\mathbb{P}_\nu(R_\delta \geq N) \geq 1/2$ for*

$$N = \min \left\{ \frac{\ln \frac{T^\star(\mu)}{T_{\min}}}{\ln \left( \left( \ln \frac{T^\star(\mu)}{T_{\min}} \right)^2 \max\{e, C\} \right)}, \frac{1}{2\delta + c} \right\}$$

*with $C = 1 + 4\gamma \log(\frac{1}{\delta}) \left(1 + \sqrt{\frac{T^\star(\mu)\Delta^2}{2\sigma^2}}\right)^2$ and $\Delta = \max_i |\mu_i - y|$.*

*Proof.* Fix some ($\sigma$-Gaussian) instance $\nu^0 = \nu$ of complexity $T^\star(\mu) = T_0 \in (T_{\min}, T_{\max})$.

For some $\zeta \in (0,1)$ to be fixed later, define the instance of mean $\mu^{n+1} = \zeta^{-1/2}\mu^n + (1 - \zeta^{-1/2})y$. We then have $\zeta^n T^\star(\mu^0) = T^\star(\mu^n)$ by hypothesis. We can thus construct a sequence of instances of length $N$ as long as $\zeta^N > \frac{T_{\min}}{T^\star(\mu^0)}$.

$$\frac{(\mu_i^{N-n-1} - \mu_i^{N-n})^2}{2\sigma^2} = \frac{(\zeta^{-(N-n-1)/2} - \zeta^{-(N-n)/2})^2(\mu_i^0 - y)^2}{2\sigma^2}$$

$$\leq \frac{\zeta^{n-N}\Delta^2}{2\sigma^2}(1 - \zeta^{1/2})^2$$

We apply Theorem 2.1 on the reversed sequence $(\nu_{N-i})_{0 \leq i \leq N}$:

$$\mathbb{P}_\nu(R_\delta > N) \geq 1 - N(2\delta + c) - \sqrt{\frac{\gamma \log(1/\delta)}{2(\zeta^{-1} - 1)}} \times \sum_{i=0}^{N-1} \left[ 1 + \sqrt{T^\star(\mu) \sup_{w \in \Delta_K} \sum_{i \in [K]} w_i \frac{\Delta^2}{2\sigma^2}(1 - \zeta^{1/2})^2} \right]$$

$$\geq 1 - N \left( 2\delta + c + \sqrt{\frac{\gamma \log(1/\delta)}{2(\zeta^{-1} - 1)}} \left( 1 + \sqrt{\frac{T^\star(\mu^0)\Delta^2}{2\sigma^2}} \right) \right)$$

$$\geq 5/8 - N(2\delta + c)$$

for

$$\zeta = \left( 1 + 4N^2\gamma \log\left(\frac{1}{\delta}\right) \left( 1 + \sqrt{\frac{T^\star(\mu^0)\Delta^2}{2\sigma^2}} \right)^2 \right)^{-1}$$

We can apply Lemma A.5 with $\rho = \frac{T^\star(\mu^0)}{T_{\min}}$, $k = 1$, $b = 0$, and $a = 4\gamma \log(1/\delta) \left( 1 + \sqrt{\frac{T^\star(\mu^0)\Delta^2}{2\sigma^2}} \right)^2$. We get that a sufficient condition is

$$N \leq \frac{\ln \frac{T^\star(\mu^0)}{T_{\min}}}{\ln \left( \left( \ln \frac{T^\star(\mu^0)}{T_{\min}} \right)^2 \max\{e, C\} \right)} \tag{5}$$

with $C = 1 + 4\gamma \log(1/\delta) \left( 1 + \sqrt{\frac{T^\star(\mu^0)\Delta^2}{2\sigma^2}} \right)^2$.

Therefore, by picking $N$ that satisfies (5) and $N \leq \frac{1}{8(2\delta + c)}$, we have that $\mathbb{P}_\nu(R_\delta > N) > \frac{1}{2}$. $\square$

**Lemma A.8** (Batch lower bound on affine sequences, expectation constraint). *For problems on which Assumption 2.2 is satisfied; for any $\delta$-correct algorithm such that, for any Gaussian instance $\nu$ satisfying $T^\star(\mu) \in (T_{\min}, T_{\max})$ $\mathbb{E}_\nu[\tau_\delta] \leq$*

$\gamma \log(1/\delta)T^\star(\boldsymbol{\mu})$, we have for any $\sigma$-Gaussian instance $\boldsymbol{\nu}$ of complexity $T^\star(\boldsymbol{\mu}) \in (T_{\min}, T_{\max})$, for the corresponding $y \in \mathbb{R}$ given by Assumption 2.2 for $\boldsymbol{\mu}$ that $\mathbb{P}_{\boldsymbol{\nu}}(R_\delta \geq N) \geq 1/2$ for

$$N \geq \min \left\{ \frac{\ln \frac{T^\star(\boldsymbol{\mu})}{T_{\min}}}{\ln \left( \left( \ln \frac{T^\star(\boldsymbol{\mu})}{T_{\min}} \right)^2 \max\{e, C'_\delta\} \right)}, \frac{1}{3} \ln \frac{T^\star(\boldsymbol{\mu})}{T_{\min}}, \frac{1}{3\delta} \right\}$$

with $C'_\delta = \max \left\{ e, 1 + 4\gamma \log(1/\delta) \ln \frac{T^\star(\boldsymbol{\mu})}{T_{\min}} \left( 1 + \sqrt{\frac{T^\star(\boldsymbol{\mu})\Delta^2}{2\sigma^2}} \right)^2 \right\}$ and $\Delta = \max_i |\mu_i - y|$.

*Proof.* For instance $\boldsymbol{\nu}$, for some algorithm satisfying $\mathbb{E}_{\boldsymbol{\nu}}[\tau_\delta] \leq \gamma \log(1/\delta)T^\star(\boldsymbol{\mu})$, we have by the Markov inequality that $\mathbb{P}_{\boldsymbol{\nu}}(\tau_\delta \geq (\gamma/c)T^\star(\boldsymbol{\mu}) \log(1/\delta)) \leq c$. Applying Lemma A.7, $\mathbb{P}_{\boldsymbol{\nu}}(R_\delta \geq N) \geq 1/2$ for

$$N = \min \left\{ \frac{\ln \frac{T^\star(\boldsymbol{\mu})}{T_{\min}}}{\ln \left( \left( \ln \frac{T^\star(\boldsymbol{\mu})}{T_{\min}} \right)^2 \max\{e, C\} \right)}, \frac{1}{2\delta + c} \right\}$$

with $C = 1 + 4\gamma/c \log(\frac{1}{\delta}) \left( 1 + \sqrt{\frac{T^\star(\boldsymbol{\mu})\Delta^2}{2\sigma^2}} \right)^2$ and $\Delta = \max_i |\mu_i - y|$.

Choosing $c = \max \left\{ \delta, \left( \log \frac{T^\star(\boldsymbol{\mu})}{T_{\min}} \right)^{-1} \right\}$, if $\delta < \left( \log \frac{T^\star(\boldsymbol{\mu})}{T_{\min}} \right)^{-1}$, then

$$N \geq \min \left\{ \frac{\ln \frac{T^\star(\boldsymbol{\mu})}{T_{\min}}}{\ln \left( \left( \ln \frac{T^\star(\boldsymbol{\mu})}{T_{\min}} \right)^2 \max\{e, C'_\delta\} \right)}, \frac{1}{3} \ln \frac{T^\star(\boldsymbol{\mu})}{T_{\min}} \right\}$$

with $C'_\delta = \max \left\{ e, 1 + 4\gamma \log(1/\delta) \ln \frac{T^\star(\boldsymbol{\mu})}{T_{\min}} \left( 1 + \sqrt{\frac{T^\star(\boldsymbol{\mu})\Delta^2}{2\sigma^2}} \right)^2 \right\}.$

If $\delta \geq \left( \log \frac{T^\star(\boldsymbol{\mu})}{T_{\min}} \right)^{-1}$,

$$N \geq \min \left\{ \frac{\ln \frac{T^\star(\boldsymbol{\mu})}{T_{\min}}}{\ln \left( \left( \ln \frac{T^\star(\boldsymbol{\mu})}{T_{\min}} \right)^2 \max\{e, C'_\delta\} \right)}, \frac{1}{3\delta} \right\}$$

$\square$

## A.3. The top-$k$ and BAI settings

All that remains is to show that our problems satisfy Assumption 2.2. We start with top-$k$, and first give a technical result giving a simple formula for $T^\star(\boldsymbol{\mu})$.

**Lemma A.9.** *For any $\boldsymbol{w} \in \Sigma_K$,*

$$\inf_{\boldsymbol{\lambda} \in Alt_{\boldsymbol{\mu}}} \left( \sum_{i \in [K]} w_i d(\mu_i, \lambda_i) \right) = \min_{\substack{b \geq k+1 \\ a \leq k}} w_a d(\mu_a, \mu_{ab}) + w_b d(\mu_b, \mu_{ab})$$

*where $\mu_{ab} = \frac{w_a \mu_a + w_b \mu_b}{w_a + w_b}$ (arms are assumed to be ordered, $\mu_1 \geq \mu_2 \geq \dots$).*

**Lemma A.10.** *In the top-k problem, setting $\boldsymbol{\mu}' = x\boldsymbol{\mu} + (1-x)\boldsymbol{y}$ where $\boldsymbol{y}$ is a constant vector and $x > 0$, $Alt_{\boldsymbol{\mu}'} = Alt_{\boldsymbol{\mu}}$, $\Delta_i^{\boldsymbol{\mu}'} = x\Delta_i^{\boldsymbol{\mu}}$ and $(T^\star(\boldsymbol{\mu}'))^{-1} = x^2(T^\star(\boldsymbol{\mu}))^{-1}$.*

*Proof.* First of all, for any two arms $i, j$, $\mu_i' - \mu_j' = x(\mu_i - \mu_j)$ with $x > 0$. Therefore, the ordering of arms is conserved, and $Alt_{\boldsymbol{\mu}'} = Alt_{\boldsymbol{\mu}}$. Moreover, since $\Delta_i^{\boldsymbol{\mu}'} = \mu_i' - \mu_{k+1}' = x(\mu_i - \mu_{k+1}) = x\Delta_i^{\boldsymbol{\mu}}$ for $i \leq k$ and $\Delta_i^{\boldsymbol{\mu}'} = \mu_k' - \mu_i' = x\Delta_i^{\boldsymbol{\mu}}$ otherwise, we do have $\Delta_i^{\boldsymbol{\mu}'} = x\Delta_i^{\boldsymbol{\mu}}$.

Furthermore,

$$(T^{\star}(\boldsymbol{\mu}'))^{-1} = \sup_{w \in \Sigma_K} \inf_{\boldsymbol{\lambda} \in Alt_{\boldsymbol{\mu}}} \left( \sum_{i \in [K]} w_i \frac{(\mu_i' - \lambda_i)^2}{2\sigma^2} \right)$$

$$= \sup_{w \in \Sigma_K} \min_{\substack{b \geq k+1 \\ a \leq k}} w_a \frac{(\mu_a' - \mu_{ab}')^2}{2\sigma^2} + w_b \frac{(\mu_b' - \mu_{ab}')^2}{2\sigma^2}$$

with

$$\mu_{ab}'(w) = \frac{w_a \mu_a' + w_b \mu_b'}{w_a + w_b} = x\mu_{ab} + (1 - x)y$$

So that

$$(T^{\star}(\boldsymbol{\mu}'))^{-1} = x^2 \sup_{w \in \Sigma_K} \min_{\substack{b \geq k+1 \\ a \leq k}} w_a \frac{(\mu_a - \mu_{ab})^2}{2\sigma^2} + w_b \frac{(\mu_b - \mu_{ab})^2}{2\sigma^2}$$

$$= x^2 (T^{\star}(\boldsymbol{\mu}))^{-1}$$

□

With these results, we can apply Lemmas A.7 and A.8. We see that the value of $y$ does not impact the proof: we thus choose the value that minimizes $\max_i |\mu_i - y|$, which is $y = \frac{\max_i \mu_i + \min_i \mu_i}{2}$.

### A.4. The thresholding setting

**Lemma A.11.** *In the thresholding bandit problem, setting $\boldsymbol{\mu}' = x\boldsymbol{\mu} + (1 - x)\boldsymbol{\tau}$ where $\boldsymbol{\tau}$ is the constant vector of value $\tau$ the threshold and $x > 0$, $Alt_{\boldsymbol{\mu}'} = Alt_{\boldsymbol{\mu}}$, $\Delta_i^{\boldsymbol{\mu}'} = x\Delta_i^{\boldsymbol{\mu}}$ and $(T^{\star}(\boldsymbol{\mu}'))^{-1} = x^2 (T^{\star}(\boldsymbol{\mu}))^{-1}$.*

*Proof.* First of all, for any arm $i$, $\mu_i' - \tau = x(\mu_i - \tau)$ with $x > 0$. Therefore, $Alt_{\boldsymbol{\mu}} = Alt_{\boldsymbol{\mu}'}$. Moreover, $\Delta_i^{\boldsymbol{\mu}'} = |\mu_i' - \tau| = x|\mu_i - \tau| = x\Delta_i^{\boldsymbol{\mu}'}$.

Furthermore,

$$(T^{\star}(\boldsymbol{\mu}'))^{-1} = \sup_{w \in \Sigma_K} \inf_{\boldsymbol{\lambda} \in Alt_{\boldsymbol{\mu}}} \left( \sum_{i \in [K]} w_i \frac{(\mu_i' - \lambda_i)^2}{2\sigma^2} \right)$$

$$= \sup_{w \in \Sigma_K} \sup_{i \in [K]} w_i \frac{(\mu_i' - \tau)^2}{2\sigma^2}$$

$$= x^2 \sup_{w \in \Sigma_K} \sup_{i \in [K]} w_i \frac{(\mu_i - \tau)^2}{2\sigma^2}$$

$$= x^2 (T^{\star}(\boldsymbol{\mu}))^{-1}$$

□

## B. Concentration and threshold for the stopping rule

We suppose that each arm is sampled once during the first $K$ time steps.

**Theorem B.1.** *Suppose that the arm distributions are $\sigma^2$-sub-Gaussian. Let $\hat\mu_{t,k}$ be the average of arm $k$ at time $t$ and $N_{t,k}$ be the number of times arm $k$ is sampled up to time $t$. With probability $1 - \delta$, for all $t > K$,*

$$\frac{1}{2}\sum_{k=1}^{K} N_{t,k}\frac{(\hat\mu_{t,k} - \mu_k)^2}{2\sigma^2} \le \frac{K}{2}\overline{W}\left(2\ln\left(\frac{e\pi^2}{6}\right) + \frac{2}{K}\ln\left(\prod_{k=1}^{K}(1 + \ln N_{t,k})^2\right) + \frac{2}{K}\ln\frac{1}{\delta}\right).$$

### B.1. Proof of the concentration theorem

We can assume *w.l.o.g.* that $\mu_k = 0$ for all $k$ and $\sigma^2 = 1$.

Let $S_{t,k} = \sum_{s=1}^{t} X_{s,k}\mathbb{I}\{k_s = k\}$. We want a bound on $\frac{1}{2}\sum_{k=1}^{K}\frac{S_{t,k}^2}{N_{t,k}}$.

We first remark that $\frac{1}{2}x^2 = \sup_\lambda \lambda x - \frac{1}{2}\lambda^2$. Apply that to $x = S_{t,k}/\sqrt{N_{t,k}}$ to get

$$\sum_{k=1}^{K}\frac{1}{2}\frac{S_{t,k}^2}{N_{t,k}} = \sup_{\lambda_1,\dots,\lambda_K}\sum_{k=1}^{K}\left(\lambda_k S_{t,k} - \frac{1}{2}N_{t,k}\lambda_k^2\right)$$

$$= \sup_{\lambda_1,\dots,\lambda_K}\sum_{s=1}^{t}\lambda_{k_s}X_{s,k_s} - \frac{1}{2}\lambda_{k_s}^2.$$

The advantage of that formulation is that we can concentrate the sum for any fixed value of $\lambda$ (or any distribution on $\lambda$) thanks to a martingale argument.

**Lemma B.2.** *For all $\rho \in \mathcal{P}(\mathbb{R}^K)$, the process $t \mapsto \mathbb{E}_{\lambda\sim\rho}\left[\exp\left(\sum_{s=1}^{t}\lambda_{k_s}X_{s,k_s} - \frac{1}{2}\lambda_{k_s}^2\right)\right]$ is a non-negative supermartingale with expectation bounded by 1.*

**Corollary B.3.** *For all $\rho \in \mathcal{P}(\mathbb{R}^K)$ and $x \ge 0$,*

$$\mathbb{P}\left(\exists t,\ \ln\mathbb{E}_{\lambda\sim\rho}\left[\exp\left(\sum_{s=1}^{t}\lambda_{k_s}X_{s,k_s} - \frac{1}{2}\lambda_{k_s}^2\right)\right] \ge x\right) \le e^{-x}.$$

*Equivalently, for all $\delta \in (0, 1]$,*

$$\mathbb{P}\left(\exists t,\ \ln\mathbb{E}_{\lambda\sim\rho}\left[\exp\left(\sum_{s=1}^{t}\lambda_{k_s}X_{s,k_s} - \frac{1}{2}\lambda_{k_s}^2\right)\right] \ge \ln\frac{1}{\delta}\right) \le \delta.$$

*Proof.* Use Ville's inequality and the fact that the process is a non-negative supermartingale. $\qquad\square$

We don't want to bound an integral over $\lambda \sim \rho$, but the supremum over $\lambda$, so we need to relate the two quantities. We do that for Gaussian priors over $\lambda$.

**Lemma B.4.** *For $\rho = \mathcal{N}(0, \mathrm{diag}(\sigma_k^{-2}))$,*

$$\ln\mathbb{E}_{\lambda\sim\rho}\left[\exp\left(\sum_{s=1}^{t}\lambda_{k_s}X_{s,k_s} - \frac{1}{2}\lambda_{k_s}^2\right)\right] = -\frac{1}{2}\sum_{k=1}^{K}\ln(1 + N_{t,k}\sigma_k^{-2}) + \frac{1}{2}\sum_{k=1}^{K}\frac{S_{t,k}^2}{(N_{t,k} + \sigma_k^2)}.$$

*Proof.*

$$
\mathbb{E}_{\lambda \sim \rho} \left[ \exp \left( \sum_{s=1}^{t} \lambda_{k_s} X_{s,k_s} - \frac{1}{2} \lambda_{k_s}^2 \right) \right]
$$

$$
= \prod_k \mathbb{E}_{\lambda_k \sim \mathcal{N}(0, \sigma_k^{-2})} \left[ \exp \left( \lambda_k S_{t,k} - \frac{1}{2} N_{t,k} \lambda_k^2 \right) \right]
$$

$$
= \prod_k \frac{1}{\sqrt{2\pi\sigma_k^{-2}}} \int_{\lambda_k} \exp \left( \lambda_k S_{t,k} - \frac{1}{2} N_{t,k} \lambda_k^2 - \frac{\sigma_k^2}{2} \lambda_k^2 \right) \mathrm{d}\lambda_k
$$

$$
= \prod_k \frac{1}{\sqrt{(1 + N_{t,k}\sigma_k^{-2})}} \frac{1}{\sqrt{2\pi(N_{t,k} + \sigma_k^2)^{-1}}} \int_{\lambda_k} \exp \left( -\frac{1}{2}(N_{t,k} + \sigma_k^2) \left( \lambda_k - \frac{S_{t,k}}{(N_{t,k} + \sigma_k^2)} \right)^2 + \frac{1}{2} \frac{S_{t,k}^2}{N_{t,k} + \sigma_k^2} \right) \mathrm{d}\lambda_k
$$

$$
= \prod_k \frac{1}{\sqrt{(1 + N_{t,k}\sigma_k^{-2})}} \exp \left( \frac{1}{2} \frac{S_{t,k}^2}{N_{t,k} + \sigma_k^2} \right)
$$

$\square$

**Corollary B.5.** *Let $\rho = \mathcal{N}(0, \mathrm{diag}(\sigma_k^{-2}))$, $\eta_{t,\max} = \max_k \frac{\sigma_k^2}{N_{t,k}}$ and $\eta_{t,\min} = \min_k \frac{\sigma_k^2}{N_{t,k}}$. Then*

$$
\frac{1}{2} \sum_{k=1}^{K} \frac{S_{t,k}^2}{N_{t,k}} \leq (1 + \eta_{t,\max}) \left( \ln \mathbb{E}_{\lambda \sim \rho} \left[ \exp \left( \sum_{s=1}^{t} \lambda_{k_s} X_{s,k_s} - \frac{1}{2} \lambda_{k_s}^2 \right) \right] + \frac{K}{2} \ln(1 + \eta_{t,\min}^{-1}) \right)
$$

*Proof.* Using Lemma B.4,

$$
\frac{1}{2} \sum_{k=1}^{K} \frac{S_{t,k}^2}{N_{t,k} + \sigma_k^2} = \ln \mathbb{E}_{\lambda \sim \rho} \left[ \exp \left( \sum_{s=1}^{t} \lambda_{k_s} X_{s,k_s} - \frac{1}{2} \lambda_{k_s}^2 \right) \right] + \frac{1}{2} \sum_{k=1}^{K} \ln(1 + N_{t,k}\sigma_k^{-2}) .
$$

Then

$$
\frac{1}{2} \sum_{k=1}^{K} \frac{S_{t,k}^2}{N_{t,k} + \sigma_k^2} \geq \frac{1}{2} \sum_{k=1}^{K} \frac{S_{t,k}^2}{N_{t,k}(1 + \eta_{t,\max})} = \frac{1}{1 + \eta_{t,\max}} \frac{1}{2} \sum_{k=1}^{K} \frac{S_{t,k}^2}{N_{t,k}}
$$

Finally, $N_{t,k}\sigma_k^{-2} \leq \eta_{t,\min}^{-1}$. $\square$

If $N_{t,k}$ was a known, unchanging number, we could choose $\sigma_k^2 \propto N_{t,k}$ to get $\eta_{t,\max} = \eta_{t,\min}$, and we would choose it to minimize the right hand side. The strategy to use that "known $N_{t,k}$" case even if they are random is to put geometric grids on the number of pulls of each arm, define distributions that are adapted to each cell of the grid, and combine them into a mixture of Gaussians.

Let $(\eta_{n_1,\ldots,n_K})_{n_1,\ldots,n_K \in \mathbb{N}}$ be non-negative real numbers that will be chosen later. For $i \in \mathbb{N}$, let $w_i = \frac{6}{\pi^2} \frac{1}{(i+1)^2}$. The weights $(w_i)$ satisfy $\sum_{i \in \mathbb{N}} w_i = 1$, hence also $\sum_{n_1,\ldots,n_K} (\prod_{k=1}^{K} w_{n_k}) = 1$.

Let $\rho_{n_1,\ldots,n_K} = \bigotimes_{k=1}^{K} \mathcal{N}(0, e^{-n_k}\eta_{n_1,\ldots,n_K}^{-1})$. This is a product distribution, with each marginal being a Gaussian with mean 0 and variance that depends on the number of grid cell.

With probability $1 - \delta$, for all $(n_1, \ldots, n_K) \in \mathbb{N}^K$ and all $t$,

$$
\ln \mathbb{E}_{\lambda \sim \rho_{n_1,\ldots,n_K}} \left[ \exp \left( \sum_{s=1}^{t} \lambda_{k_s} X_{s,k_s} - \frac{1}{2} \lambda_{k_s}^2 \right) \right] \leq \ln \frac{1}{\delta} + \sum_{k=1}^{K} \ln \frac{1}{w_{n_k}} .
$$

This is simply an union bound using Corollary B.3, with weight $\prod_{k=1}^{K} w_{n_k}$ for $\rho_{n_1,\ldots,n_K}$.

In particular, there exists $(n_1, \ldots, n_k)$ such that for all $k \in [K]$, $e^{n_k} \leq N_{t,k} \leq e^{n_k+1}$. For that choice, $e^{-1}\eta_{n_1,\ldots,n_K} \leq \frac{e^{n_k}\eta_{n_1,\ldots,n_K}}{N_{t,k}} \leq \eta_{n_1,\ldots,n_K}$. For those values, using Corollary B.5 with $\sigma_k^2 = e^{n_k}\eta_{n_1,\ldots,n_K}$, with probability $1 - \delta$,

$$\frac{1}{2}\sum_{k=1}^{K} \frac{S_{t,k}^2}{N_{t,k}} \leq (1 + \eta_{n_1,\ldots,n_K}) \left( \ln \frac{1}{\delta} + \sum_{k=1}^{K} \ln \frac{1}{w_{n_k}} + \frac{K}{2}\ln(1 + e\eta_{n_1,\ldots,n_K}^{-1}) \right)$$

$$\leq (1 + \eta_{n_1,\ldots,n_K}) \left( \ln \frac{e^{K/2}}{\delta} + \sum_{k=1}^{K} \ln \frac{1}{w_{n_k}} + \frac{K}{2}\ln(1 + \eta_{n_1,\ldots,n_K}^{-1}) \right)$$

$$= (1 + \eta_{n_1,\ldots,n_K}) \left( \ln \frac{(\sqrt{e}\pi^2/6)^K \prod_{k=1}^{K}(n_k+1)^2}{\delta} + \frac{K}{2}\ln(1 + \eta_{n_1,\ldots,n_K}^{-1}) \right)$$

This is where we choose $\eta_{n_1,\ldots,n_K}$ to minimize the right hand side.

By Lemma A.3 of (Degenne, 2019), the minimal value is attained at some $\eta_{n_1,\ldots,n_K}$ such that

$$(1 + \eta_{n_1,\ldots,n_K}) \left( \ln \frac{(\sqrt{e}\pi^2/6)^K \prod_{k=1}^{K}(n_k+1)^2}{\delta} + \frac{K}{2}\ln(1 + \eta_{n_1,\ldots,n_K}^{-1}) \right)$$

$$= \frac{K}{2}\overline{W}\left( 1 + \frac{2}{K}\ln \frac{(\sqrt{e}\pi^2/6)^K \prod_{k=1}^{K}(n_k+1)^2}{\delta} \right)$$

By the choice of $n_k$, it satisfies $n_k \leq \ln N_{t,k}$. We get that with probability $1 - \delta$, for all $t$,

$$\frac{1}{2}\sum_{k=1}^{K} \frac{S_{t,k}^2}{N_{t,k}} \leq \frac{K}{2}\overline{W}\left( 1 + \frac{2}{K}\ln \frac{(\sqrt{e}\pi^2/6)^K \prod_{k=1}^{K}(1 + \ln N_{t,k})^2}{\delta} \right)$$

$$= \frac{K}{2}\overline{W}\left( \frac{2}{K}\ln \left( (e\pi^2/6)^K \prod_{k=1}^{K}(1 + \ln N_{t,k})^2 \right) + \frac{2}{K}\ln \frac{1}{\delta} \right) .$$

This ends the proof of the theorem.

**B.2. Upper bounds on $\beta(t,\delta)$ and on $\gamma_r$**

We choose the threshold

$$\beta(t,\delta) = \frac{K}{2}\overline{W}\left( 2\ln\left(\frac{e\pi^2}{6}\right) + \frac{2}{K}\ln\left( \prod_{k=1}^{K}(1 + \ln N_{t,k})^2 \right) + \frac{2}{K}\ln \frac{1}{\delta} \right) .$$

We can get an upper bound that is not random by maximizing over $(N_{t,k})_{k\in[K]}$ under the constraint $\sum_{k=1}^{K} N_{t,k} = t$. We get

$$\beta(t,\delta) \leq \frac{K}{2}\overline{W}\left( 2\ln\left(\frac{e\pi^2}{6}\right) + 4\ln\left(1 + \ln \frac{t}{K}\right) + \frac{2}{K}\ln \frac{1}{\delta} \right) .$$

We can get further upper bounds by using $\overline{W}(x) \leq x + \ln x + 1/2 \leq 2x$. This gives

$$\beta(t,\delta) \leq 2K\ln\left(\frac{e\pi^2}{6}\right) + 4K\ln\left(1 + \ln \frac{t}{K}\right) + 2\ln \frac{1}{\delta}$$

$$\leq 2K\ln\left(\frac{e\pi^2}{6}\right) + 4K\ln \frac{t}{K} + 2\ln \frac{1}{\delta} .$$

The right asymptotic for $\beta(t,\delta)$ as $\delta \to 0$ is $\ln(1/\delta)$. We lost a factor 2 in the upper bound above.

**Lemma B.6.** *Let $\gamma_r$ be the solution to $\beta(\bar{t}_r, \delta) = \gamma_r$, for $\bar{t}_r = 2(Kl_{1,r}/T_0 + \gamma_r)T_r$ and $l_{1,r} = 32T_0 \ln(2\sqrt{2K}T_r)$. Then*

$$\gamma_r \le 4\ln\frac{1}{\delta} + 8K\ln(T_r) + 4K(11 + \ln K) \,.$$

*Proof.* We use an upper bound for $\beta(t, \delta)$: $\gamma_r$ is bounded from above by the solution $\gamma_r'$ to

$$\gamma = 2K\ln\left(\frac{e\pi^2}{6}\right) + 4K\ln\left(2(32\ln(2\sqrt{2K}T_r) + \frac{\gamma}{K})T_r\right) + 2\ln\frac{1}{\delta} \,.$$

Then either $\gamma_r' \le 8K\ln(2\sqrt{2K}T_r)$ or $\gamma_r'$ is less than the solution to

$$\gamma = 2K\ln\left(\frac{e\pi^2}{6}\right) + 4K\ln\left(10\frac{\gamma T_r}{K}\right) + 2\ln\frac{1}{\delta}$$

$$= 2K\ln\left(\frac{50e\pi^2}{3}\right) + 4K\ln\left(\frac{\gamma T_r}{K}\right) + 2\ln\frac{1}{\delta} \,.$$

That is,

$$\gamma_r' = 4K\overline{W}\left(\frac{1}{2K}\ln\frac{1}{\delta} + \ln(T_r) + \frac{1}{2}\ln\frac{800e\pi^2}{3}\right)$$

$$\le 4\ln\frac{1}{\delta} + 8K\ln(T_r) + 4K\ln\frac{800e\pi^2}{3} \,.$$

At this point, we get

$$\gamma_r \le \max\left\{4\ln\frac{1}{\delta} + 8K\ln(T_r) + 4K\ln\frac{800e\pi^2}{3}, 8K\ln(2\sqrt{2K}T_r)\right\}$$

$$\le 8K\ln(T_r) + \max\left\{4\ln\frac{1}{\delta} + 4K\ln\frac{800e\pi^2}{3}, 8K\ln(2\sqrt{2K})\right\}$$

$$\le 8K\ln(T_r) + 4\ln\frac{1}{\delta} + 4K\ln\frac{800e\pi^2}{3} + 8K\ln(2\sqrt{2K})$$

$$\le 8K\ln(T_r) + 4\ln\frac{1}{\delta} + 4K\ln\left(\frac{6400e\pi^2}{3}K\right)$$

$$\le 8K\ln(T_r) + 4\ln\frac{1}{\delta} + 4K(11 + \ln K) \,.$$

$\square$

## C. Proofs related to the algorithm

### C.1. Additional Lemmas

**Lemma C.1.** *Let $(X_i)_{i\in\mathbb{N}}$ be i.i.d. $\sigma^2$-sub-Gaussian random variables with mean $\mu$. For $n \in \mathbb{N}$, let $\hat{\mu}_n$ be the average of the first $n$ random variables. Then*

$$\mathbb{P}(\exists n \ge N, \ \hat{\mu}_n \ge \mu + \varepsilon) \le e^{-\frac{N\varepsilon^2}{2\sigma^2}} \,,$$

$$\mathbb{P}(\exists n \ge N, \ \hat{\mu}_n \le \mu - \varepsilon) \le e^{-\frac{N\varepsilon^2}{2\sigma^2}} \,.$$

*Proof.* Given $(X_1, \ldots, X_N)$, the process $M_n(\lambda) : n \mapsto e^{\lambda\sum_{i=1}^{N+n}(X_i-\mu) - \frac{1}{2}(N+n)\sigma^2\lambda^2}$ is a nonnegative supermartingale for any $\lambda \in \mathbb{R}$ by the sub-Gaussian hypothesis, with expectation $e^{\lambda\sum_{i=1}^{N}(X_i-\mu) - \frac{1}{2}N\sigma^2\lambda^2}$ at $n = 0$.

By Ville's inequality,

$$\mathbb{P}(\exists n, \ M_n(\lambda) \geq 1/\delta \mid X_1, \ldots, X_N) \leq \delta e^{\lambda \sum_{i=1}^{N}(X_i - \mu) - \frac{1}{2}N\sigma^2\lambda^2}$$

For all $\lambda \in \mathbb{R}$ and all $\delta \in (0, 1)$,

$$\mathbb{P}\left(\exists n \geq N, \ \lambda \sum_{i=1}^{n}(X_i - \mu) - \frac{1}{2}n\sigma^2\lambda^2 \geq \ln(1/\delta)\right) = \mathbb{E}\left[\mathbb{P}(\exists n \geq 0, \ M_n(\lambda) \geq 1/\delta \mid X_1, \ldots, X_N)\right]$$

$$\leq \delta \mathbb{E}\left[e^{\lambda \sum_{i=1}^{N}(X_i - \mu) - \frac{1}{2}N\sigma^2\lambda^2}\right]$$

$$\leq \delta.$$

Reordering, we get, for $\lambda \geq 0$,

$$\mathbb{P}\left(\exists n \geq N, \ \hat{\mu}_n \geq \mu + \frac{1}{2}\sigma^2\lambda + \frac{1}{N\lambda}\ln\frac{1}{\delta}\right) \leq \mathbb{P}\left(\exists n \geq N, \ \hat{\mu}_n \geq \mu + \frac{1}{2}\sigma^2\lambda + \frac{1}{n\lambda}\ln\frac{1}{\delta}\right)$$

$$\leq \delta.$$

Choose $\delta = e^{-\frac{N\varepsilon^2}{2\sigma^2}}$ and $\lambda = \frac{\varepsilon}{\sigma^2}$ to obtain

$$\mathbb{P}\left(\exists n \geq N, \ \hat{\mu}_n \geq \mu + \varepsilon\right) \leq e^{-\frac{N\varepsilon^2}{2\sigma^2}}.$$

The second inequality is obtained similarly, with $\lambda \leq 0$. $\qquad \square$

**Lemma C.2.** *The probability of $\mathcal{E}_r$ satisfies*

$$\mathbb{P}(\mathcal{E}_r) \geq 1 - 2K \exp(-2^r l_{1,r} \varepsilon_r^2 / 2\sigma^2).$$

*Proof.* $\mathcal{E}_r$ is the event that $\|\boldsymbol{\mu} - \tilde{\boldsymbol{\mu}}^r\|_\infty \leq \varepsilon_r$ and $\|\boldsymbol{\mu} - \hat{\boldsymbol{\mu}}^r\|_\infty \leq \varepsilon_r$. We use an union bound over the arms to bound the probability of the complement $\mathcal{E}_r^c$. For each $i \in [K]$, $\tilde{\mu}_i^r$ and $\hat{\mu}_i^r$ are empirical means of at least $2^r l_{1,r}$ samples. We can thus apply Lemma C.1 (twice, once for deviations from above and once for deviations from below). $\qquad \square$

**Lemma C.3.** *Let $p_r \in (0, 1]$. For the choice $\varepsilon_r = \sqrt{\frac{2\sigma^2}{2^r l_1} \ln \frac{2K}{p_r}}$, the probability of the event $\mathcal{E}_r$ is $\mathbb{P}(\mathcal{E}_r) \geq 1 - p_r$.*

### C.2. Proof of Lemma 3.7

If $\overline{T}^\star(\hat{B}_r) > T_r$ then the algorithm does not enter the second batch of the phase by definition of the algorithm.

If $\overline{T}^\star(\hat{B}_r) \leq T_r$ then by the choice of $\gamma_r$ and Lemma 3.4, under $\mathcal{E}_r$ the stopping condition is triggered.

We now prove the complexity upper bounds. Let $\mathcal{C}_r$ be the event that the algorithm attains phase $r$ and does not stop at that phase. We proved that $\{\overline{T}^\star(\hat{B}_r) \leq T_r\} \cap \mathcal{E}_r \subseteq \mathcal{C}_r^c$ for all $r$. That is, $\mathcal{C}_r \subseteq \mathcal{E}_r^c \cup \{\overline{T}^\star(\hat{B}_r) > T_r\}$.

Recall that $R^* = \min\{r \mid \forall r' \geq r, \ \mathcal{E}_{r'} \implies \overline{T}^\star(\hat{B}_{r'}) \leq T_{r'}\}$.

$$R_\delta = \sum_{r=1}^{+\infty} \mathbb{I}(\mathcal{C}_{r-1}) + \mathbb{I}(\mathcal{C}_{r-1} \wedge \{\overline{T}^\star(\hat{B}_r) \leq T_r\}).$$

$$\leq R^* + 2 \sum_{r=R^*+1}^{+\infty} \mathbb{I}(\mathcal{C}_{r-1}) + \sum_{r=1}^{R^*} \mathbb{I}(\mathcal{C}_{r-1} \wedge \{\overline{T}^\star(\hat{B}_r) \leq T_r\}).$$

By definition of $R^*$, for $r \geq R^*$ we have $\{\overline{T}^\star(\hat{B}_r) > T_r\} \subseteq \mathcal{E}_r^c$. Using that property and the inclusion we proved on $\mathcal{C}_r$ we have, for $r > R^*$,

$$\mathcal{C}_{r-1} \subseteq \mathcal{E}_{r-1}^c \cup \{\overline{T}^\star(\hat{B}_{r-1}) > T_{r-1}\} \subseteq \mathcal{E}_{r-1}^c.$$

Therefore,

$$R_\delta \leq R^* + 2 \sum_{r=R^*+1}^{+\infty} \mathbb{I}(\mathcal{E}_{r-1}^c) + \sum_{r=1}^{R^*} \mathbb{I}(\mathcal{C}_{r-1} \wedge \{\overline{T}^*(\hat{B}_r) \leq T_r\}) \,.$$

When $\mathcal{E}_r$ happens $T^\star(\boldsymbol{\mu}) \leq \overline{T}^*(\hat{B}_r)$, hence $\{\overline{T}^*(\hat{B}_r) \leq T_r\} \subseteq \mathcal{E}_r^c \cup \{T^\star(\boldsymbol{\mu}) \leq T_r\}$.

$$R_\delta \leq R^* + 2 \sum_{r=R^*+1}^{+\infty} \mathbb{I}(\mathcal{E}_{r-1}^c) + \sum_{r=1}^{R^*} \mathbb{I}(\mathcal{E}_r^c) + \sum_{r=1}^{R^*} \mathbb{I}(\{T^\star(\boldsymbol{\mu}) \leq T_r\})$$

$$\leq R^* + 1 + 2 \sum_{r=1}^{+\infty} \mathbb{I}(\mathcal{E}_r^c) + \sum_{r=1}^{R^*} \mathbb{I}(\{T^\star(\boldsymbol{\mu}) \leq T_r\}) \,.$$

Finally, $\sum_{r=1}^{R^*} \mathbb{I}(\{T^\star(\boldsymbol{\mu}) \leq T_r\}) = \max\left\{0, R^* - \left\lceil \log_2 \frac{T^\star(\boldsymbol{\mu})}{T_0} \right\rceil_+ \right\}$, with $\lceil x \rceil_+ = \max\{0, \lceil x \rceil\}$.

We now bound the sample complexity $\tau_\delta$. Since $\bar{t}_r$ is an upper bound on the sample complexity up to phase $r$,

$$\tau_\delta \leq \sum_{r=1}^{+\infty} \bar{t}_r \mathbb{I}\{C_{r-1}\}$$

$$\leq \bar{t}_{R^*} + \sum_{r=R^*+1}^{+\infty} \bar{t}_r \mathbb{I}\{C_{r-1}\}$$

$$\leq \bar{t}_{R^*} + \sum_{r=R^*+1}^{+\infty} \bar{t}_r \mathbb{I}\{E_{r-1}^c\} \,.$$

### C.3. Proof of the batch and complexity upper bounds

We finish the proof of Theorem 3.11 from where we stopped in the main text. We have

$$\mathbb{E}[R_\delta] \leq r^* + \max\left\{0, r^* - \left\lceil \log_2 \frac{T^\star(\boldsymbol{\mu})}{T_0} \right\rceil_+ \right\} + 1 + 2 \sum_{r=1}^{+\infty} p_r \,,$$

$$\mathbb{E}[\tau_\delta] \leq \bar{t}_{r^*} + \sum_{r=1}^{+\infty} p_r \bar{t}_{r+1} \,.$$

In those expressions, $r^* = \max\{r_0, r_1\}$ with $r_0 = \min\{r \mid 2\varepsilon_r \leq b(\boldsymbol{\mu})\}$, $r_1 = \min\{r \mid T_r \geq eT^\star(\boldsymbol{\mu})\}$, and $\bar{t}_r = (Kl_{1,r}/T_0 + 2\gamma_r)T_r$ with $l_{1,r}/T_0 = 32\ln(2\sqrt{2KT_r})$.

The choice of $p_r$ is a trade-off between the sums and $r_0$. We choose $p_r = T_{r+1}^2$.

**Bounding the sums** The sum in the batch complexity is bounded by $T_0^{-2}/3$. The sum that appears in the sample complexity is

$$\sum_{r=\max\{r_0,r_1\}+1}^{+\infty} p_{r-1} \bar{t}_r = \sum_{r=\max\{r_0,r_1\}+1}^{+\infty} \frac{\bar{t}_r}{T_r^2} \,.$$

We will need the values of a few sums.

$$\sum_{r=1}^{+\infty} \frac{1}{T_r} = \sum_{r=1}^{+\infty} \frac{1}{2^r T_0} = \frac{1}{T_0} \,,$$

$$\sum_{r=1}^{+\infty} \frac{\ln T_r}{T_r} = \frac{1}{T_0} \sum_{r=1}^{+\infty} \frac{r\ln 2 + \ln T_0}{2^r} = \frac{\ln(4T_0)}{T_0} \,.$$

Let $c_{K,\delta} = 4\ln\frac{1}{\delta} + 4K(11 + \ln K)$. By Lemma B.6, $\gamma_r \leq 8K\ln(T_r) + c_{K,\delta}$. An upper bound on the sample complexity until the end of phase $r$ is then

$$\begin{aligned}
\bar{t}_r &= (Kl_{1,r}/T_0 + 2\gamma_r)T_r \\
&= (32K\ln(2\sqrt{2K}T_r) + 2\gamma_r)T_r \\
&\leq (48K\ln(T_r) + 32K\ln(2\sqrt{2K}) + c_{K,\delta})T_r .
\end{aligned}$$

The sum that appears in the sample complexity is at most

$$\sum_{r=1}^{+\infty} \frac{\bar{t}_r}{T_r^2} \leq \frac{48K\ln(4T_0) + 32K\ln(2\sqrt{2K}) + c_{K,\delta}}{T_0} .$$

**Bound on $r^*$ and $\bar{t}_{r^*}$**

$$\bar{t}_{r^*} \leq (48K\ln(\max\{T_{r_0}, T_{r_1}\}) + 32K\ln(2\sqrt{2K}) + c_{K,\delta})\max\{T_{r_0}, T_{r_1}\} .$$

Recall that $r_0 = \min\{r \geq 0 \mid 2\varepsilon_r \leq b(\boldsymbol{\mu})\}$, $r_1 = \min\{r \geq 0 \mid T_r \geq eT^\star(\boldsymbol{\mu})\}$.

If we get an upper bound $n$ on $T_i$, we then have $r_i \leq \log_2 \frac{n}{T_0}$. We get a bound on $T_{r_1}$ from its definition: if $r_0 \geq 1$, then $T_{r_1 - 1} \leq eT^*(\boldsymbol{\mu})$ hence in the end $T_{r_1} \leq \max\{T_0, 2eT^*(\boldsymbol{\mu})\}$.

If $r_0 \geq 1$, then since $\varepsilon_{r_0 - 1} \geq b(\boldsymbol{\mu})/2$, we get an inequality on $T_{r_0 - 1}$.

$$\sqrt{\frac{2\sigma^2}{2^{r_0 - 1}l_{1, r_0 - 1}}\ln\left(2KT_{r_0}^2\right)} \geq \frac{b(\boldsymbol{\mu})}{2} .$$

With the value of $l_{1,r}$ and using $2T_{r_0 - 1} = T_{r_0}$, this becomes in the end

$$T_{r_0} \leq \max\{T_0, \frac{\sigma^2}{b(\boldsymbol{\mu})^2}\} .$$

Let $T_b^\star(\boldsymbol{\mu}) = \max\{\frac{\sigma^2}{b(\boldsymbol{\mu})^2}, 2eT^*(\boldsymbol{\mu})\}$. We have proved $\max\{T_{r_0}, T_{r_1}\} \leq \max\{T_0, T_b^\star(\boldsymbol{\mu})\}$, hence

$$\bar{t}_{r^*} \leq (48K\ln\max\{T_0, T_b^\star(\boldsymbol{\mu})\} + 32K\ln(2\sqrt{2K}) + c_{K,\delta})\max\{T_0, T_b^\star(\boldsymbol{\mu})\}$$

and $r^* \leq \max\left\{0, \log_2 \frac{T_b^\star(\boldsymbol{\mu})}{T_0}\right\}$.

**Putting things together**   For the round complexity,

$$\mathbb{E}\left[R_\delta\right] \leq \max\left\{0, \log_2 \frac{T_b^\star(\boldsymbol{\mu})}{T_0}\right\} + \max\left\{0, \max\left\{0, \log_2 \frac{T_b^\star(\boldsymbol{\mu})}{T_0}\right\} - \left\lceil \log_2 \frac{T^\star(\boldsymbol{\mu})}{T_0}\right\rceil_+\right\} + 1 + T_0^{-2}$$

If $T_b^\star(\boldsymbol{\mu}) \geq T_0$, then

$$\begin{aligned}
\mathbb{E}\left[R_\delta\right] &\leq 2\log_2 \frac{T_b^\star(\boldsymbol{\mu})}{T_0} - \log_2 \frac{T^\star(\boldsymbol{\mu})}{T_0} + 1 + T_0^{-2} \\
&\leq \log_2 \frac{T_b^\star(\boldsymbol{\mu})}{T_0} + \log_2 \frac{T_b^\star(\boldsymbol{\mu})}{T^\star(\boldsymbol{\mu})} + 2
\end{aligned}$$

If $T_b^\star(\boldsymbol{\mu}) < T_0$, by definition, $T_0 > T_b^\star(\boldsymbol{\mu}) \geq T^\star(\boldsymbol{\mu})$. We then have

$$\begin{aligned}
\mathbb{E}\left[R_\delta\right] &\leq 0 + \max\left\{0, \log_2 \frac{T_b^\star(\boldsymbol{\mu})}{T_0} - 0\right\} + 1 + T_0^{-2} \\
&\leq 1 + T_0^{-2}
\end{aligned}$$

So that finally,

$$\mathbb{E}\left[R_\delta\right] \le \max\left\{\log_2 \frac{T_b^\star(\boldsymbol{\mu})}{T_0} + \log_2 \frac{T_b^\star(\boldsymbol{\mu})}{T^\star(\boldsymbol{\mu})}, 0\right\} + 1 + T_0^{-2} \,,$$

$$\mathbb{E}\left[\tau_\delta\right] \le (48K \ln \max\{T_0, T_b^\star(\boldsymbol{\mu})\} + 32K \ln(2\sqrt{2K}) + c_{K,\delta}) \max\{T_0, T_b^\star(\boldsymbol{\mu})\}$$
$$+ \frac{48K \ln(4T_0) + 32K \ln(2\sqrt{2K}) + c_{K,\delta}}{T_0} \,.$$

Let us simplify the sample complexity.

$$32K \ln(2\sqrt{2K}) + c_{K,\delta} = 32K \ln(2\sqrt{2K}) + 4 \ln \frac{1}{\delta} + 4K(11 + \ln K)$$

$$= 4 \ln \frac{1}{\delta} + 4K(5 \ln K + 11 + 4\ln(8))$$

$$\le 4 \ln \frac{1}{\delta} + 20K(\ln K + 4) \,.$$

$$\mathbb{E}\left[\tau_\delta\right] \le (48K \ln \max\{T_0, T_b^\star(\boldsymbol{\mu})\} + 4 \ln \frac{1}{\delta} + 20K(\ln K + 4)) \max\{T_0, T_b^\star(\boldsymbol{\mu})\}$$

$$+ (48K \ln(4T_0) + 4 \ln \frac{1}{\delta} + 20K(\ln K + 4))T_0^{-1}$$

$$= \left(4 \ln\left(\frac{1}{\delta}\right) + 20K(\ln K + 4)\right)(\max\{T_0, T_b^\star(\boldsymbol{\mu})\} + T_0^{-1})$$

$$+ 48K(\max\{T_0, T_b^\star(\boldsymbol{\mu})\} \ln \max\{T_0, T_b^\star(\boldsymbol{\mu})\} + T_0^{-1} \ln(4T_0)) \,.$$

### C.4. Implication between the two assumptions

We prove Lemma 3.10.

**Lemma C.4.** *Let $\boldsymbol{\nu}$ and $\boldsymbol{\nu}'$ be two instances and let $\omega_{\boldsymbol{\mu}} = \arg\max_\omega \inf_{\lambda \in Alt_{\boldsymbol{\mu}}} \sum_{i=1}^K \omega_i(\mu_i - \lambda_i)^2$. Then*

$$\sqrt{T^\star(\boldsymbol{\mu})^{-1}} - \sqrt{T^\star(\boldsymbol{\mu}')^{-1}} \le \frac{1}{\sqrt{2\sigma^2}} \|\mu - \mu'\|_\infty \,.$$

*Proof.* For $\omega \in \Sigma_K$ and $\boldsymbol{x} \in \mathbb{R}^K$, let $\|\boldsymbol{x}\|_\omega = \sqrt{\sum_{i=1}^K \omega_i x_i^2}$. It satisfies the triangle inequality and $\|\boldsymbol{x}\|_\omega \le \|\boldsymbol{x}\|_\infty$. For any $\boldsymbol{\lambda}$ and $\omega$,

$$\|\boldsymbol{\mu} - \boldsymbol{\lambda}\|_\omega \le \|\boldsymbol{\mu}' - \boldsymbol{\lambda}\|_\omega + \|\boldsymbol{\mu} - \boldsymbol{\mu}'\|_\omega \,.$$

We can take an infimum on both sides over lambda in $Alt_{\boldsymbol{\mu}}$ and then apply the result to $\omega_{\boldsymbol{\mu}}$ to get

$$\sqrt{2\sigma^2 T^\star(\boldsymbol{\mu})^{-1}} \le \inf_{\boldsymbol{\lambda} \in Alt_{\boldsymbol{\mu}}} \|\boldsymbol{\mu}' - \boldsymbol{\lambda}\|_{\omega_{\boldsymbol{\mu}}} + \|\boldsymbol{\mu} - \boldsymbol{\mu}'\|_{\omega_{\boldsymbol{\mu}}} \,.$$

Either $Alt_{\boldsymbol{\mu}} = Alt_{\boldsymbol{\mu}'}$ and we can replace by that on the right hand side, or $\boldsymbol{\mu}' \in Alt_{\boldsymbol{\mu}}$. In that second case $\inf_{\boldsymbol{\lambda} \in Alt_{\boldsymbol{\mu}}} \|\boldsymbol{\mu}' - \boldsymbol{\lambda}\|_{\omega_{\boldsymbol{\mu}}} = 0 \le \inf_{\boldsymbol{\lambda} \in Alt_{\boldsymbol{\mu}'}} \|\boldsymbol{\mu}' - \boldsymbol{\lambda}\|_{\omega_{\boldsymbol{\mu}}}$. We thus have

$$\sqrt{2\sigma^2 T^\star(\boldsymbol{\mu})^{-1}} \le \inf_{\boldsymbol{\lambda} \in Alt_{\boldsymbol{\mu}'}} \|\boldsymbol{\mu}' - \boldsymbol{\lambda}\|_{\omega_{\boldsymbol{\mu}}} + \|\boldsymbol{\mu} - \boldsymbol{\mu}'\|_{\omega_{\boldsymbol{\mu}}} \,.$$

We maximize over $\omega$ to get $\inf_{\boldsymbol{\lambda} \in Alt_{\boldsymbol{\mu}'}} \|\boldsymbol{\mu}' - \boldsymbol{\lambda}\|_{\omega_{\boldsymbol{\mu}}} \le \sqrt{2\sigma^2 T^\star(\boldsymbol{\mu}')^{-1}}$, hence

$$\sqrt{2\sigma^2 T^\star(\boldsymbol{\mu})^{-1}} \le \sqrt{2\sigma^2 T^\star(\boldsymbol{\mu}')^{-1}} + \|\boldsymbol{\mu} - \boldsymbol{\mu}'\|_{\omega_{\boldsymbol{\mu}}}$$

$$\le \sqrt{2\sigma^2 T^\star(\boldsymbol{\mu}')^{-1}} + \|\boldsymbol{\mu} - \boldsymbol{\mu}'\|_\infty \,.$$

After dividing by $\sqrt{2\sigma^2}$, this is the inequality of the lemma. $\qquad\square$

**Corollary C.5.** *For all $\boldsymbol{\mu}$ and $\boldsymbol{\mu}'$,*

$$\ln T^\star(\boldsymbol{\mu}') - \ln T^\star(\boldsymbol{\mu}) \leq \sqrt{\frac{2}{\sigma^2} T^\star(\boldsymbol{\mu}')} \, \|\boldsymbol{\mu} - \boldsymbol{\mu}'\|_\infty \,.$$

*Proof.*

$$\begin{aligned}
\ln T^\star(\boldsymbol{\mu}') - \ln T^\star(\boldsymbol{\mu}) &= 2\ln\left(1 + \frac{\sqrt{T^\star(\boldsymbol{\mu})^{-1}} - \sqrt{T^\star(\boldsymbol{\mu}')^{-1}}}{\sqrt{T^\star(\boldsymbol{\mu}')^{-1}}}\right) \\
&\leq 2\frac{\sqrt{T^\star(\boldsymbol{\mu})^{-1}} - \sqrt{T^\star(\boldsymbol{\mu}')^{-1}}}{\sqrt{T^\star(\boldsymbol{\mu}')^{-1}}} \\
&\leq \sqrt{\frac{2}{\sigma^2} T^\star(\boldsymbol{\mu}')} \, \|\boldsymbol{\mu} - \boldsymbol{\mu}'\|_\infty \,.
\end{aligned}$$

$\square$

**Corollary C.6.** *For all $\boldsymbol{\mu}$ and $\boldsymbol{\mu}'$ with $\|\boldsymbol{\mu} - \boldsymbol{\mu}'\|_\infty \leq \sqrt{\sigma^2/(2T^\star(\boldsymbol{\mu}))}$,*

$$|\ln T^\star(\boldsymbol{\mu}') - \ln T^\star(\boldsymbol{\mu})| \leq \sqrt{\frac{8}{\sigma^2} T^\star(\boldsymbol{\mu})} \, \|\boldsymbol{\mu} - \boldsymbol{\mu}'\|_\infty \,.$$

*Proof.* One of the two inequalities we need to prove is due to Corollary C.5. For the other, by the same corollary,

$$\ln T^\star(\boldsymbol{\mu}') - \ln T^\star(\boldsymbol{\mu}) \leq \sqrt{\frac{2}{\sigma^2} T^\star(\boldsymbol{\mu}')} \, \|\boldsymbol{\mu} - \boldsymbol{\mu}'\|_\infty \,.$$

It remains to show $T^\star(\boldsymbol{\mu}') \leq 4T^\star(\boldsymbol{\mu})$. By Lemma C.4 and the the hypothesis on $\|\boldsymbol{\mu} - \boldsymbol{\mu}'\|_\infty$,

$$\sqrt{T^\star(\boldsymbol{\mu})^{-1}} - \sqrt{T^\star(\boldsymbol{\mu}')^{-1}} \leq \frac{1}{\sqrt{2\sigma^2}}\|\boldsymbol{\mu} - \boldsymbol{\mu}'\|_\infty \leq \frac{1}{2}\sqrt{T^\star(\boldsymbol{\mu})^{-1}} \,.$$

Reordering proves the inequality. $\square$

### C.5. Proofs for Top-K and thresholding bandits

This section is devoted to the proof of Lemma 3.12. We start with a preliminary result allowing us to compute $\overline{w}^\star(B)$ once we have found a suitable instance in $B$.

**Lemma C.7.** *If all instances in $B$ share the same correct answer $i^\star$ and if there exists some mean vector $\boldsymbol{b} \in B$ such that*

$$\inf_{\boldsymbol{\nu} \in B} \inf_{\boldsymbol{\lambda} \in Alt_{\boldsymbol{b}}} \sum_i w_i(\boldsymbol{b}) \frac{(\mu_i - \lambda_i)^2}{2\sigma^2} \geq \inf_{\boldsymbol{\lambda} \in Alt_{\boldsymbol{b}}} \sum_i w_i(\boldsymbol{b}) \frac{(b_i - \lambda_i)^2}{2\sigma^2} \tag{6}$$

*where $w(\boldsymbol{\mu}) = \operatorname{argmax}_{w \in \Sigma_K} \inf_{\boldsymbol{\lambda} \in Alt_{\boldsymbol{\mu}}} \sum_i w_i \frac{(\mu_i - \lambda_i)^2}{2\sigma^2}$, then $\overline{T}^\star(B) = \max_{\boldsymbol{\mu} \in B} T^\star(\boldsymbol{\mu})$.*

*Proof.* For some $w \in \Sigma_K$, writing $f(w, \boldsymbol{\mu}') = \inf_{\boldsymbol{\lambda} \in Alt_{\boldsymbol{b}}} \sum_i w_i \frac{(\mu_i' - \lambda_i)^2}{2\sigma^2}$ for clarity,

$$\begin{aligned}
\inf_{\boldsymbol{\nu} \in \hat{B}_r} f(w, \boldsymbol{\mu}) &\leq f(w, \boldsymbol{b}) && \text{because } \boldsymbol{b} \in \hat{B}_r \\
&\leq f(w_{\boldsymbol{b}}, \boldsymbol{b}) && \text{from the definition of } w_{\boldsymbol{b}} \\
&\leq \inf_{\boldsymbol{\mu}' \in \hat{B}_r} f(w_{\boldsymbol{b}}, \boldsymbol{\mu}') && \text{from the hypothesis}
\end{aligned}$$

so that $w^\star(\boldsymbol{b}) = \operatorname{argmax}_{w \in \Sigma_K} \inf_{\boldsymbol{\nu} \in \hat{B}_r} f(\boldsymbol{w}, \nu) = \overline{w}^\star(B)$.

$$\overline{T}^\star(B) = \left( \inf_{\boldsymbol{\nu}' \in B} \inf_{\boldsymbol{\lambda} \in Alt_{\boldsymbol{\nu}'}} \sum_i \overline{w}_i^\star(B) \frac{(\mu_i' - \lambda_i)^2}{2\sigma^2} \right)^{-1}$$

$$= \left( \inf_{\boldsymbol{\nu}' \in B} \inf_{\boldsymbol{\lambda} \in Alt_{\boldsymbol{\nu}'}} \sum_i w_i^\star(\boldsymbol{b}) \frac{(\mu_i' - \lambda_i)^2}{2\sigma^2} \right)^{-1}$$

$$= \left( \inf_{\boldsymbol{\lambda} \in Alt_{\boldsymbol{b}}} \sum_i w_i^\star(\boldsymbol{b}) \frac{(b_i - \lambda_i)^2}{2\sigma^2} \right)^{-1} \qquad \text{by Equation (6)}$$

$$= T^\star(\boldsymbol{b})$$

hence $\overline{T}(B) \leq \max_{\boldsymbol{\nu} \in B} T^\star(\boldsymbol{\nu})$. By definition, we have the other inequality, and we conclude. $\qquad \square$

**Lemma 3.12.** *In Top-$k$, including best arm identification, as well as for thresholding bandits, Assumption 3.9 holds. That is, for all $\boldsymbol{\mu}$ and all $\varepsilon \geq 0$,*

$$\overline{T}^\star(\mathcal{B}_\infty(\boldsymbol{\mu}, \varepsilon)) = \max_{\boldsymbol{\mu}' \in \mathcal{B}_\infty(\boldsymbol{\mu}, \varepsilon)} T^\star(\boldsymbol{\mu}') \,.$$

We prove the result separately for top-$k$ and TBP. In both cases, we give a certain mean vector $\boldsymbol{b}$, and then we show it satisfies the premise of Lemma C.7, then we use that result to show that $\overline{T}^\star(\mathcal{B}_\infty(\boldsymbol{\mu}, \varepsilon)) = T^\star(\boldsymbol{b})$.

*Proof of Lemma 3.12 for top-$k$.* Assume without loss of generality that the arms are well ordered, $\mu_1 \geq \mu_2 \geq \cdots \geq \mu_K$.

If $\mu_k - \mu_{k+1} \leq 2\varepsilon$, then there exists $\boldsymbol{b} \in \mathcal{B}_\infty(\boldsymbol{\mu}, \varepsilon)$ such that $b_k = b_{k+1}$. $\overline{T}^\star(\mathcal{B}_\infty(\boldsymbol{\mu}, \varepsilon)) \geq \max_{\boldsymbol{\nu}' \in \mathcal{B}_\infty(\boldsymbol{\mu}, \varepsilon)} T^\star(\boldsymbol{\nu}') \geq T^\star(\boldsymbol{b}) = +\infty$, therefore

$$\overline{T}^\star(\mathcal{B}_\infty(\boldsymbol{\mu}, \varepsilon)) = \max_{\boldsymbol{\nu}' \in \mathcal{B}_\infty(\boldsymbol{\mu}, \varepsilon)} T^\star(\boldsymbol{\nu}') \,.$$

When $\mu_k - \mu_{k+1} > 2\varepsilon$, define

$$\begin{cases} b_i = \mu_i - \varepsilon & \text{if } i \leq k \\ b_i = \mu_i + \varepsilon & \text{if } i \geq k+1 \end{cases}$$

and, for any $\boldsymbol{\nu}'$,

$$w_{\boldsymbol{\nu}'} = \operatorname*{argmax}_{w \in \Sigma_K} \inf_{\boldsymbol{\lambda} \in Alt_{\boldsymbol{\nu}'}} \sum_i w_i \frac{(\mu_i' - \lambda_i)^2}{2\sigma^2} \,.$$

Let there be some $\boldsymbol{\nu}' \in \mathcal{B}_\infty(\boldsymbol{\mu}, \varepsilon)$. Then for $i \leq k$, $\mu_i' \geq \mu_i - \varepsilon = b_i$ and for $i \geq l+1$, $\mu_i' \leq \mu_i + \varepsilon = b_i$, and $Alt_{\boldsymbol{\nu}'} = Alt_{\boldsymbol{b}}$. We know from Lemma A.9

$$\min_{\boldsymbol{\lambda} \in Alt_{\boldsymbol{b}}} \sum_i w_i(\boldsymbol{b}) \frac{(\mu_i' - \lambda_i)^2}{2\sigma^2} = w_a(\boldsymbol{b}) \frac{(\mu_a' - \mu_{aj}')^2}{2\sigma^2} + w_j(\boldsymbol{b}) \frac{(\mu_j' - \mu_{aj}')^2}{2\sigma^2} \qquad (7)$$

for some $a \leq k < k+1 \leq j$, and $\mu_{aj}' = \frac{w_a(\boldsymbol{b})}{w_a(\boldsymbol{b}) + w_j(\boldsymbol{b})} \mu_a' + \frac{w_j(\boldsymbol{b})}{w_a(\boldsymbol{b}) + w_j(\boldsymbol{b})} \mu_j'$.

- If $\mu_{aj}' \in (b_a, \mu_a')$, then

$$w_a(\boldsymbol{b})(\mu_a' - \mu_{aj}')^2 + w_j(\boldsymbol{b})(\mu_j' - \mu_{aj}')^2 \geq 0 + w_j(\boldsymbol{b})(\mu_j' - b_a)^2$$
$$\geq w_a(\boldsymbol{b})(b_a - b_a)^2 + w_j(\boldsymbol{b})(b_j - b_a)^2$$

- If $\mu_{aj}' \in (b_j, b_a)$,

$$w_a(\boldsymbol{b})(\mu_a' - \mu_{aj}')^2 + w_j(\boldsymbol{b})(\mu_j' - \mu_{aj}')^2 \geq w_a(\boldsymbol{b})(b_a - \mu_{aj}')^2 + w_j(\boldsymbol{b})(b_j - \mu_{aj}')^2$$

- If $\mu'_{aj} \in (\mu'_j, b_j)$,

$$w_a(\boldsymbol{b})(\mu'_a - \mu'_{aj})^2 + w_j(\boldsymbol{b})(\mu'_j - \mu'_{aj})^2 \geq w_a(\boldsymbol{b})(\mu'_a - b_j)^2 + 0$$
$$\geq w_a(\boldsymbol{b})(b_a - b_j)^2 + w_j(\boldsymbol{b})(b_j - b_j)^2$$

In all three cases,

$$w_a(\boldsymbol{b})(\mu'_a - \mu'_{aj})^2 + w_j(\boldsymbol{b})(\mu'_j - \mu'_{aj})^2 \geq \inf_{\lambda \in [b_j, b_a]} w_a(\boldsymbol{b})(b_a - \lambda)^2 + w_j(\boldsymbol{b})(b_j - \lambda)^2$$
$$\geq \inf_{\boldsymbol{\lambda} \in Alt_{\boldsymbol{b}}} \sum_i w_i(\boldsymbol{b})(b_i - \lambda_i)^2$$

and therefore, by Equation (7),

$$\forall \boldsymbol{\nu}' \in B_\infty(\boldsymbol{\mu}, \varepsilon), \ \inf_{\boldsymbol{\lambda} \in Alt_{\boldsymbol{b}}} \sum_i w_i(\boldsymbol{b}) \frac{(\nu_i - \lambda_i)^2}{2\sigma^2} \geq \inf_{\boldsymbol{\lambda} \in Alt_{\boldsymbol{b}}} \sum_i w_i(\boldsymbol{b}) \frac{(b_i - \lambda_i)^2}{2\sigma^2}$$

and therefore

$$\inf_{\boldsymbol{\nu} \in \mathcal{B}_\infty(\boldsymbol{\mu}, \varepsilon)} \inf_{\boldsymbol{\lambda} \in Alt_{\boldsymbol{b}}} \sum_i w_i(\boldsymbol{b}) \frac{(\nu_i - \lambda_i)^2}{2\sigma^2} \geq \inf_{\boldsymbol{\lambda} \in Alt_{\boldsymbol{b}}} \sum_i w_i(\boldsymbol{b}) \frac{(b_i - \lambda_i)^2}{2\sigma^2} \tag{8}$$

We can thus apply Lemma C.7, and conclude

$$\overline{T}(\mathcal{B}_\infty(\boldsymbol{\mu}, \varepsilon)) = \max_{\boldsymbol{\nu} \in \mathcal{B}_\infty(\boldsymbol{\mu}, \varepsilon)} T^\star(\boldsymbol{\nu}) \,.$$

$\square$

*Proof of Lemma 3.12 for thresholding bandits.* If for some $i$, $|\mu_i - \tau| \leq \varepsilon$, then there exists $\boldsymbol{b} \in \mathcal{B}_\infty(\boldsymbol{\mu}, \varepsilon)$ such that $b_i = \tau$. Therefore, $\overline{T}^\star(\mathcal{B}_\infty(\boldsymbol{\mu}, \varepsilon)) \geq \max_{\boldsymbol{\nu}' \in \mathcal{B}_\infty(\boldsymbol{\mu}, \varepsilon)} T^\star(\boldsymbol{\nu}') \geq T^\star(\boldsymbol{b}) = +\infty$, therefore

$$\overline{T}^\star(\mathcal{B}_\infty(\boldsymbol{\mu}, \varepsilon)) = \max_{\boldsymbol{\nu}' \in \mathcal{B}_\infty(\boldsymbol{\mu}, \varepsilon)} T^\star(\boldsymbol{\nu}') \,.$$

When $\min_k |\mu_k - \tau| > \varepsilon$, define $U = \{i \in [K] : \mu_i > \tau\}$ and $L = [K] \setminus U$. Define

$$\begin{cases} b_i = \mu_i - \varepsilon & \text{if } i \in U \\ b_i = \mu_i + \varepsilon & \text{if } i \in L. \end{cases}$$

and for any $\boldsymbol{\nu}'$, $w_{\boldsymbol{\nu}'} = \text{argmax}_{w \in \Sigma_K} \inf_{\boldsymbol{\lambda} \in Alt_{\boldsymbol{\nu}'}} \sum_i w_i \frac{(\mu'_i - \lambda_i)^2}{2\sigma^2}$.

Let there be some $\boldsymbol{\nu} \in \mathcal{B}_\infty(\boldsymbol{\mu}, \varepsilon)$. We know $\min_{\boldsymbol{\lambda} \in Alt_{\boldsymbol{b}}} \sum_i w_i(\boldsymbol{b}) \frac{(\mu'_i - \lambda_i)^2}{2\sigma^2} = w_j(\boldsymbol{b}) \frac{(\mu'_j - \tau)^2}{2\sigma^2}$ for some $j$.

For all $i \in U$, $\mu'_i \geq \mu_i - \varepsilon = b_i > \tau$; for all $i \in L$, $\mu'_i \leq \mu_i + \varepsilon = b_i < \tau$. Therefore,

$$w_j(\boldsymbol{b}) \frac{(\mu'_j - \tau)^2}{2\sigma^2} \geq w_j(\boldsymbol{b}) \frac{(b_j - \tau)^2}{2\sigma^2} \geq \inf_{\boldsymbol{\lambda} \in Alt_{\boldsymbol{b}}} \sum_i w_i(\boldsymbol{b}) \frac{(b_i - \lambda_i)^2}{2\sigma^2}$$

We thus have $\forall \boldsymbol{\nu}' \in \mathcal{B}_\infty(\boldsymbol{\mu}, \varepsilon)$, $\inf_{\boldsymbol{\lambda} \in Alt_{\boldsymbol{b}}} \sum_i w_i(\boldsymbol{b}) \frac{(\mu'_i - \lambda_i)^2}{2\sigma^2} \geq \inf_{\boldsymbol{\lambda} \in Alt_{\boldsymbol{b}}} \sum_i w_i(\boldsymbol{b}) \frac{(b_i - \lambda_i)^2}{2\sigma^2}$, and

$$\inf_{\boldsymbol{\nu} \in \hat{B}_r} \inf_{\boldsymbol{\lambda} \in Alt_{\boldsymbol{b}}} \sum_i w_i(\boldsymbol{b}) \frac{(\nu_i - \lambda_i)^2}{2\sigma^2} \geq \inf_{\boldsymbol{\lambda} \in Alt_{\boldsymbol{b}}} \sum_i w_i(\boldsymbol{b}) \frac{(b_i - \lambda_i)^2}{2\sigma^2}$$

and by Lemma C.7,

$$\overline{T}(\mathcal{B}_\infty(\boldsymbol{\mu}, \varepsilon)) = \max_{\boldsymbol{\nu} \in \mathcal{B}_\infty(\boldsymbol{\mu}, \varepsilon)} T^\star(\boldsymbol{\nu}) \,.$$

$\square$

