# OpenReview forum: "The Batch Complexity of Bandit Pure Exploration"
_ICML.cc/2025/Conference — ICML 2025 poster_

### Official Review · Reviewer_KpQ9 · 2025-03-06

**Overall Recommendation:** 4

**Summary:**

The authors derive an instance-dependent lower bound on the batch complexity required by any $\delta$-correct pure exploration algorithm. This lower bound is expressed as a function of the instance’s complexity,  $T^\star(\mu)$, and shows that as sample efficiency improves, the minimal number of batches required increases. This result is novel in that it quantifies, for every instance, the minimal “round” or batch count needed.

 Building on the Track-and-Stop methodology, the paper introduces a new algorithm called Phased Explore then Track (PET). The algorithm is designed in phases. In each phase, it first performs uniform exploration to reliably estimate the means and the instance-dependent complexity; then, if a predetermined complexity threshold is met, it transitions into a more adaptive sampling phase. The authors prove that PET is $\delta$-correct and provide upper bounds on both its sample complexity (which is nearly optimal, up to logarithmic factors in  $1/\delta$ ) and its batch complexity.

The general results are then specialized to well-studied pure exploration tasks such as Best-Arm Identification (BAI), Top-k, and thresholding bandits. For these settings, the authors show that the algorithm’s performance nearly matches the derived lower bounds.
		Empirical results are provided for both BAI and thresholding bandit problems. The results indicate that PET achieves a favorable balance between sample and batch complexities by using only a few batches while keeping the sample complexity near-optimal.
		For the stopping rule as they consider non-parametric distribution, they use mixtures method.

**Claims And Evidence:**

Theorem 2.3 provides a lower bound on the necessary number of batches; Theorem 3.5 establishes the $\delta$-correctness of the PET algorithm; and Theorem 3.11 gives upper bounds on both the expected batch and sample complexities.

All the proofs are given in appendix, and seem correct.

Some numerical experiments support the theoretical findings.


## update after rebuttal"
I have read the discussions and finally I keep my assessment unchanged

**Essential References Not Discussed:**

nothing special

**Experimental Designs Or Analyses:**

no relevant here

**Methods And Evaluation Criteria:**

The synthetic numerical experiments provide sufficient evidence given the theoretical nature of the work.

**Other Comments Or Suggestions:**

On page~1, the introduction could benefit from clearer phrasing in the following sentence: "...samples is fixed and the objective is to minimize the probability of error, \textbf{we} talk about fixed-budget pure exploration (Audibert et al., 2010)." As the paper focuses on fixed-confidence settings, the use of the word \textbf{we} here may cause confusion.

 A clearer motivation for investigating batched methods would enhance the paper. Fully adaptive algorithms, such as Track-and-Stop, require solving a min-max optimization problem at each round to determine optimal sampling proportions, which can be computationally intensive and impractical in many applications. In contrast, batched methods fix the sampling strategy for an entire batch, thereby reducing computational demands and enabling parallel sampling. Emphasizing this computational trade-off would further justify the focus on batched approaches, particularly in scenarios with delayed feedback or limited adaptivity.

 In the paper "Optimal $\delta$-Correct Best-Arm Selection for Heavy-Tailed Distributions," a batch algorithm is introduced. The authors should cite this work and discuss the differences between their approach and the one presented in that paper.

Regarding Assumption 3.8 and line 5 of the algorithm, the paper employs an $l$-norm to quantify the distance between instances. Given that KL divergence often provides a tighter and more informative measure—especially in Gaussian settings—could the authors clarify the rationale behind this choice? Would it be possible or beneficial to replace the $l$-norm with a KL divergence measure in these parts of the analysis? Clarifying this point would help readers understand the trade-offs between tractability and the precision of the bounds.

 The definition of $\mu^0$ as the initial instance appears to be missing. Including this definition would enhance clarity.


 It would be helpful to add some intuition for Assumption 2.2, such as: "It essentially states that the ‘difficulty’ (as measured by the inverse of the complexity) scales as $x^2$ when the instance is linearly scaled in this manner. This property is useful for constructing sequences of instances and analyzing how the complexity evolves as the instances become progressively more challenging (i.e., as the gaps shrink)."

On page~7, it seems that the authors intended to refer to Lemma~3.7 rather than Theorem~3.7, as the latter does not appear in the paper. Clarifying this reference would improve accuracy.

**Other Strengths And Weaknesses:**

The use of batches in best-arm identification problems is real, and interesting to study theoretically.
The contributions are solid and rather clear, though pretty unsurprising.
This work hence seems to be a decent incremental contribution.

**Questions For Authors:**

Is it possible to directly recover the results of Garivier–Kaufmann (2016) by imposing some structure on the batches? Clarification on this point would be appreciated.

**Relation To Broader Scientific Literature:**

The state of the art is correct

**Theoretical Claims:**

proof coherence checked globally

---

> ### Author Rebuttal · Authors · 2025-04-01
>
> We thank the reviewer for their insightful suggestions.
>
> The suggested paper, ``Optimal $\delta$-correct best arm selection for heavy tailed distributions'', contains a BAI algorithm for a different class of non-parametric distributions (given by a moment constraint).
> They also use batches in a Track-and-Stop method. The main motivation for the batches seems to be to reduce the computational cost of the algorithm, and minimizing the expected batch complexity is not a goal they consider (and no bound is given).
> They mostly use constant batch sizes, which would lead to a batch complexity that is proportional to $\log(1/\delta)$, which should not be the case according to the lower bound.
> Thank you for the reference: we will include discussion of that paper in our revision.
>
> On the usage of an $l$-norm instead of a KL divergence, it is first due to our goal of tackling the non-parametric sub-Gaussian setting, in which this is the natural distance measure.
> If we were to consider parametric families of distribution, we would want to use a KL divergence.
> However a hurdle with that lies in Lemma 3.10. Indeed, we would need to have guarantees about the variation of the KL in a confidence region around a mean estimate.
> Investigating how to get past this problem and generalize the results for $\tilde{T}^*$ instead of $T^*$ would be interesting future work.
>
> We do not understand the last question about recovering the results of Garivier and Kaufmann 2016 by imposing structure on the batches. Could you expand on what you mean by that question?
> Their algorithm is not batched (or rather has all batches of size 1), so we don't see what the structure on the batches would be.

---

> > ### Comment · Reviewer_KpQ9 · 2025-04-01
> >
> > >  We do not understand the last question about recovering the results of Garivier and Kaufmann 2016 by imposing structure on the batches. Could you expand on what you mean by that question? Their algorithm is not batched (or rather has all batches of size 1), so we don't see what the structure on the batches would be.
> >
> > The question was: when you allow the number of rounds to be as large as wanted, does the complexity you obtain by your analysis tend to T^*(\mu) ? but I agree that this is not so much the goal of your contribution.

---

> > > ### Author Response · Authors · 2025-04-08
> > >
> > > Thank you for your clarification. It is possible to have a main term (in $\log(1/\delta)$) for the sample complexity $(1+\varepsilon)T^*(\mu)\ln(1/\delta)$ for $\varepsilon > 0$ as small as we want with a few changes: multiplying the length of each subsequent phase by $1+\alpha$ with very small $\alpha$ instead of by 2; changing the definition of $l_1$ to reflect this change, and multiplying it by a constant; and multiplying the probability $p_r$ that appears in $\varepsilon_r$ by $\alpha$ as well.
> > > However, both the batch complexity and the lower order terms of the sample complexity become bigger with those changes.

---

### Official Review · Reviewer_mSwP · 2025-03-10

**Overall Recommendation:** 4

**Summary:**

The paper investigates the batch complexity of fixed-confidence pure exploration problems in stochastic multi-armed bandits (MAB), where the algorithm is allowed to change its sampling strategy only across batches. The authors derive novel instance-dependent lower bounds on the number of batches required by any sample-efficient algorithm for general pure exploration tasks. They propose a general batched algorithm named "Phased Explore then Track" (PET), inspired by the Track-and-Stop method, and provide upper bounds on its expected sample and batch complexities. Numerical results are presented to show the effectivenes of the algorithm.

**Claims And Evidence:**

The claims presented in the submission are supported by convincing evidence. The authors provide theoretical proofs for the lower bounds on batch complexity (Lemma 2.1 and Theorem 2.3), as well as detailed analyses of their proposed algorithm's upper bounds. Experimental results validate the effectiveness of the proposed algorithm

**Essential References Not Discussed:**

The most relevant references have been discussed.

**Experimental Designs Or Analyses:**

No significant issues were identified with these experimental designs or analyses.

**Methods And Evaluation Criteria:**

The proposed methods and evaluation criteria are appropriate for the problem at hand.

**Other Comments Or Suggestions:**

Typos:

Page 4 line 177: "or" → "of"

Page 5 line 329: "an other assumption" → "another assumption"

Page 8 line 439: "That effect that was first observed" → "That effect was first observed"

**Other Strengths And Weaknesses:**

See comments in the other sections.

**Questions For Authors:**

1. How sensitive is the algorithm's performance to the choice of $T_0$? How significantly does it affect your empirical results?

2. Could you elaborate further on potential approaches to replace uniform exploration (Lines 6-9 in Algorithm 1) with more adaptive strategies for general pure exploration problems? Would such approaches significantly reduce practical sample complexities?

3. Have you considered computational aspects when implementing your algorithm? In practice, how costly is it to $\bar{w}^\star$ and $\bar{T}^\star(\cdot)$? Is there a way to solve it through bisection methods similarly to (Garivier & Kaufmann, 2016) for the Top-1 (BAI) setting?

**Relation To Broader Scientific Literature:**

The paper's contributions relate to existing literature on batched bandit algorithms and pure exploration problems and collaborative bandits with communications (which can be interpreted as batched observation). It builds upon fundamental work on fixed-confidence pure exploration (Garivier & Kaufmann, 2016). It generalizes previous studies that focused specifically on BAI or Top-k identification to the class of pure exploration problems with batched observation.

**Theoretical Claims:**

I skimmed through the proofs of the main theoretical claims. The proofs appear sound and mathematically rigorous, with no evident errors.

---

> ### Author Rebuttal · Authors · 2025-04-01
>
> We are thankful to the reviewer for their helpful comments.
>
> Firstly, we'd like to redirect the reviewer to the two last paragraphs of our response to reviewer yvDd: the bound on the sample complexity in Theorem 3.11 was not accurate in the regime of big $T_0$. The parameter $T_0$ impacts the size of the first batch, and needs to strike a balance between sample optimality for easy instances, and round complexity for hard instances.
> We have not tried varying $T_0$ in the experiments, to try and keep all the compared algorithms with the same initial batch size so none would be particularly penalized by how easy/hard the tested instance was.
>
> We have thought of a few ways to replace uniform exploration. One would be to try and generalize BAI arm elimination schemes (see for example Algorithm 2 of Jin et al 2023), in which suboptimal arms stop being sampled. In general pure exploration, one could imagine stopping an arm $i$ once $N_i\geq \overline{w}^*_i(B)\overline{T}^*(B)$, since no instance in the confidence set would require more samples of that arm.
> We could also forgo uniform sampling altogether, only sampling according to $\overline{w}^*(B)\overline{T}^*(B)$ at each batch. However, either of those methods would make the confidence set $B$ behave in more complex ways.
>
> In the most general setting we consider, there is no obvious way of computing $\overline{w}^*(B)$ and $\overline{T}^*(B)$. Under Assumption 3.9, one would need to compute $w^*(\mu)$ and $T^*(\mu)$ over $B$ and find the supremum of $T^*$ (heuristically, finding the most difficult instance in $B$).
> However, in BAI and thresholding this is much easier, since we know precisely where the hardest instance $b$ lies.
> In BAI, it is the instance in which the best arm is moved down as much as possible, and the other arms moved up as much as possible.
> In thresholding, it is the instance in which every arm is moved closer to the threshold. See Appendix C.5 for the proofs, and line 1330 for definition of $b$ in the BAI case.
> Because of this, computing $\overline{w}^*(B)$ and $\overline{T}^*(B)$ has exactly the same cost as computing $w^*(\hat{\mu}_t)$ and $T^*(\hat{\mu}_t)$ in Track-and-Stop, and any method used to compute those also applies to our algorithm.

---

### Official Review · Reviewer_yvDd · 2025-03-13

**Overall Recommendation:** 3

**Summary:**

This work studies the batch complexity in the bandit pure exploration including best arm identification, top-k arm identification, and thresholding bandit problems. The paper begins by establishing a lower bound on the number of batches required for an algorithm that is $\delta$-correct for any Gaussian instance with sample complexity in a certain range. It then proposes an algorithm, named PET, that works in phase. Theoretical analysis shows that its batch complexity is close to the lower bound, and its sample complexity is also asymptotically optimal in the high confidence regime.

**Claims And Evidence:**

This paper is highly technical, not easy to read. I am not confident about my understanding of the claims made in this paper.

**Essential References Not Discussed:**

None.

**Experimental Designs Or Analyses:**

- I believe some experiments on Top-K or TBP should also have been conducted.
- In addition, I hope to see some experiments supporting the statement “$T_{min}$ is a tradeoff between prior knowledge and batch complexity”.

**Methods And Evaluation Criteria:**

The suggested algorithm is sensible to me. The sample complexity and the batch complexity are key performance metrics for batched pure exploration problems.

**Other Comments Or Suggestions:**

I would like to recommend the authors to refine the statements for better readability. For example,
- In Assumption 2.2, it is totally unclear whether $y$ is a vector or a scalar. I would rather write: For any mean vector $\mu$, there exists a vector $y$ such that $T^\star(x \mu + (1-x) y) = x^{-2} T^\star(\mu)$ for all $x \in [0,1]$
-  Lemma 2.1 is also extremely cryptic. Are $T_{min}$ and $T_{max}$ specific to the algorithm? Should it be $S_N$ instead of $S_n$?
-  In Algorithm 1, the variables $r$, $t$, $N_i$ are initialized but never updated.
- …

**Other Strengths And Weaknesses:**

A clear weakness of this paper is readability. See my suggestions below.

**Questions For Authors:**

Although I was not able to follow all the details, I am curious about the meaning of $T_0$ in Algorithm 1. It seems like an algorithm parameter, but it appears in Theorem 3.11 only, which implies that a larger $T_0$ leads to smaller batch complexity and smaller sample complexity without losing anything. Does Theorem 3.11 hold only if $T^\star(\mu) \geq T_0$? Otherwise, why don't we set $T_0$ to be extremely large.

**Relation To Broader Scientific Literature:**

I believe the research question that this paper asks is interesting and impactful. Given that the batched queries can be thought of as a restriction in the adaptivity, the results in this paper suggest how much we can reduce the adaptivity while preserving the sample complexity. I would believe that high level insights can also be applied in different learning tasks.

**Theoretical Claims:**

I had a hard time parsing and understanding the statements in the main body. I gently believe that the proofs are solid, but I am not sure.

---

> ### Author Rebuttal · Authors · 2025-04-01
>
> We are grateful for the reviewer's insights and corrections.
>
> Thank you for pointing out a few unclear passages.
> In Assumption 2.2, $y$ (in normal font) is a scalar, and $\bf{y}$ (in bold) is the vector $(y,y,...,y)$.
> We will make sure to clarify this.
> In Algorithm 1, we indeed did not detail where $r$, $t$ and $N_i$ (respectively the round count, the timestep count, and the number of samples of arm $i$) are updated.
> We will add a line to explicitly mention their update.
>
> In Lemma 2.1, $T_{\min}$ and $T_{\max}$ are indeed specific to the algorithm - and to the degree of optimality considered, impacting the values of $\gamma$ and $c$ - all algorithms are not uniformly optimal everywhere. While $T_{\max}$ could be set to $+\infty$ for some algorithms, fixing some $T_{\min}$ is necessary to guarantee the finiteness of $\gamma$, as explained around line 209.
> No algorithm can achieve a sample complexity that is withing a constant multiple of $T^*$ if $T^* \approx 10^{-10}$, as every arm must be sampled at least once: $T_{\min}$ should be at least of order 1.
> That Lemma should also indeed use $S_N$ instead of $S_n$, thank you for noticing that typo.
>
> It is correct that $T_0$ should have more of an impact on the bound of Theorem 3.11.
> This bound is indeed only valid for $T_b^*(\mu)$ big enough compared to $T_0$.
> More precisely, in the Appendix, line 1173, $r_0$ and $r_1$ are necessarily strictly positive, which we forgot to reflect in the final bound.
> To account for the case of large $T_0$, the batch complexity bound should be modified to be the maximum of 4 and the expression in Theorem 3.11.
> In the sample complexity bound, one needs to replace every instance of $T_b^*(\mu)$ with $\max\{ T_b^*(\mu),32KT_0\ln(2\sqrt{2K}T_0)\}$.
> We are very thankful to the reviewer for pointing out this mistake.
>
> With that correction, one can see the need to balance $T_0$: a very small $T_0$ will lead to a small initial batch size, so we can efficiently solve even easy instance without wasting sample complexity, but at the cost of way more batches for complex instances. A very big $T_0$ will minimize the number of batches required, but the initial batch might be way bigger than necessary, requiring more samples than would be optimal.

---

### Official Review · Reviewer_mFdv · 2025-03-15

**Overall Recommendation:** 4

**Summary:**

The paper investigates pure exploration problems with a specific focus on batch complexity.

First, the authors establish a theoretical lower bound for batch complexity and characterize it in relation to sample complexity, which is well understood from previous work. They then propose the PET algorithm, which nearly attains the lower bound.

Both theoretical analysis and experimental results demonstrate the effectiveness of PET.

## update after rebuttal
Overall, I appreciate the technical novelty of this paper and will therefore maintain my current score.

**Claims And Evidence:**

Yes.

**Essential References Not Discussed:**

I am not aware of any missing prior works.

**Experimental Designs Or Analyses:**

The design of conducted experiments are seems sound.

**Methods And Evaluation Criteria:**

The work is mostly theoretical but also contain some experiments which have reasonable design.

**Other Comments Or Suggestions:**

It would be better to restructure Subsection 3.1. Perhaps defining the threshold first and then presenting Lemma 3.1 would improve clarity.

Line 117: "is or order" → "is order".

**Other Strengths And Weaknesses:**

Strenghts:
* The authors address three well-known problems—BAI, Top-K, and TBP—within a unified framework, with their approach and results applicable to all.
* Considering the problem for sub-Gaussian distributions is an interesting contribution, as most previous work has focused on the 1-parameter exponential family.
* The paper provides rigorous theoretical results along with experiments demonstrating the effectiveness of the proposed algorithm.

**Questions For Authors:**

1. Can you elaborate on the inequality in Line 94? The explanation provided is quite brief.
2. I did not fully understand the reasoning behind Lemma 3.1. You assume a sub-Gaussian distribution and then use similar quantity from the Gaussian distribution (GLR), along with a new threshold, to derive a concentration inequality. Is this inequality a novel result? I am unsure whether the cited previous work has addressed this specific case. Could you clarify?

**Relation To Broader Scientific Literature:**

The key contributions of the paper are in the area of online decision-making (bandits).

**Theoretical Claims:**

I checked the flow of some parts of proofs but I’m not certain all are correct.

---

> ### Author Rebuttal · Authors · 2025-04-01
>
> We thank the reviewer for their helpful suggestions.
>
> The inequality on line 94 comes from Donsker-Varadhan duality, and can more precisely be found in Lemma 2 of Wang 2021, upper bounding the difference of means by the square root of the KL divergence for two sub-Gaussian variables.
>
> Lemma 3.1 is a version of the Chernoff stopping rule (relaxed with the same inequality as above), which can be found for BAI in Garivier and Kaufmann 2016.
> One can find a generalization of the Chernoff stopping rule for any pure exploration problem for example in Theorem 1 of Degenne et al. 2019 (Non-asymptotic Pure Exploration by Solving Games).
> Our concentration result is in Lemma 3.2. While this specific threshold beta is a new result, it employs the classic techniques used to derive such Chernoff stopping rule thresholds (we were surprised not to find such a threshold for the sub-Gaussian case in the literature).

---

### Decision · Program_Chairs · 2025-05-01

**Decision:**

Accept (poster)

**Comment:**

**Summary:** This work studies the batch complexity in pure exploration, proposing a novel characterization that differs from the standard $\log(1/\Delta_2)$ bound. Instead, it argues that the optimal batch complexity should scale as $\log(T^*/T_{\min})$, where $T^*$ denotes the optimal sample complexity and $T_{\min}$ is a given lower bound. The authors analyze several variants of the multi-armed bandit problem, establishing both upper and lower bounds on the batch complexity that nearly match.

**Justification:** All reviewers agree that this paper contributes significantly to the study of pure exploration, particularly in the underexplored area of batch complexity. Notably, it provides the first known lower bound for batch complexity in this setting. However, the current bounds on several key quantities, such as batch and sample complexity, remain somewhat loose. This reduces the perceived strength of the proposed framework and algorithms. I recommend that the authors include further discussion or refinements to better clarify the tightness of these bounds.